# ConceptAttention: Diffusion Transformers Learn Highly Interpretable Features

Alec Helbling [1]    Tuna Han Salih Meral [2]    Benjamin Hoover [1 3]    Pinar Yanardag [2]    Duen Horng (Polo) Chau [1]

## Abstract

Do the rich representations of multi-modal diffusion transformers (DiTs) exhibit unique properties that enhance their interpretability? We introduce CONCEPTATTENTION, a novel method that leverages the expressive power of DiT attention layers to generate high-quality saliency maps that precisely locate textual concepts within images. Without requiring additional training, CONCEPTATTENTION repurposes the parameters of DiT attention layers to produce highly contextualized *concept embeddings*, contributing the major discovery that performing linear projections in the output space of DiT attention layers yields significantly sharper saliency maps compared to commonly used cross-attention maps. CONCEPTATTENTION even achieves state-of-the-art performance on zero-shot image segmentation benchmarks, outperforming 15 other zero-shot interpretability methods on the ImageNet-Segmentation dataset. CONCEPTATTENTION works for popular image models and even seamlessly generalizes to video generation. Our work contributes the first evidence that the representations of multi-modal DiTs are highly transferable to vision tasks like segmentation.

## 1. Introduction

Diffusion models have recently gained widespread popularity, emerging as the state-of-the-art approach for a variety of generative tasks, particularly text-to-image synthesis (Rombach et al., 2022). These models transform random noise into photorealistic images guided by textual descriptions, achieving unprecedented fidelity and detail. Despite the impressive generative capabilities of diffusion models, our understanding of their internal mechanisms remains limited. Diffusion models operate as black boxes, where the

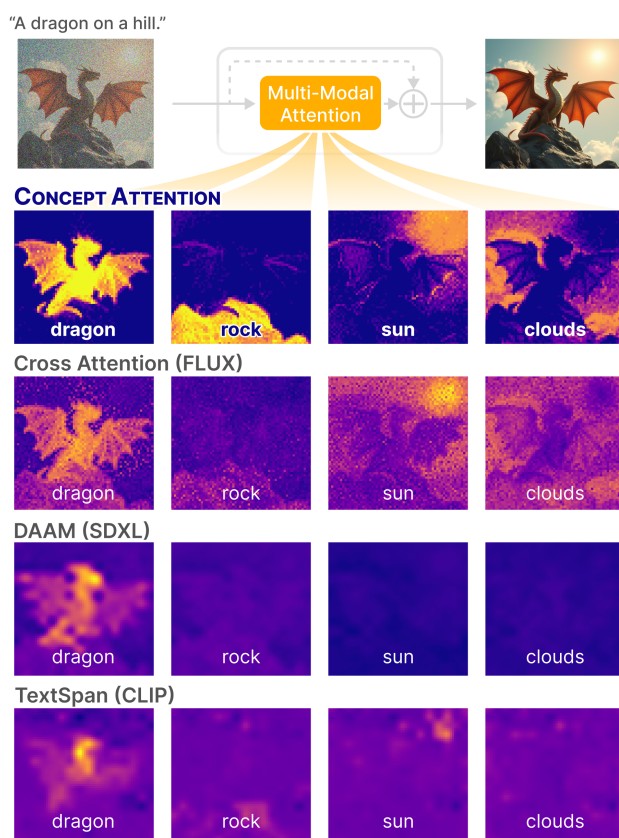

*Figure 1.* CONCEPTATTENTION **produces saliency maps that precisely localize the presence of textual concepts in images.** We compare Flux raw cross attention, DAAM (Tang et al., 2022) with SDXL, and TextSpan (Gandelsman et al., 2024) for CLIP.

relationships between input prompts and generated outputs are visible, but the decision-making processes that connect them are hidden from human understanding.

Existing work on interpreting T2I models has predominantly focused on UNet-based architectures (Podell et al., 2023; Rombach et al., 2022), which utilize shallow cross-attention mechanisms between prompt embeddings and image patch representations. UNet *cross attention maps* can produce high-fidelity saliency maps that predict the location of textual concepts (Tang et al., 2022) and have found numerous applications in tasks like image editing (Hertz et al., 2022; Chefer et al., 2023). However, the interpretability

[1]Georgia Tech [2]Virginia Tech [3]IBM Research. Correspondence to: Alec Helbling <alechelbling@gatech.edu>.

*Proceedings of the $42^{nd}$ International Conference on Machine Learning*, Vancouver, Canada. PMLR 267, 2025. Copyright 2025 by the author(s).

Code: alechelbling.com/ConceptAttention/

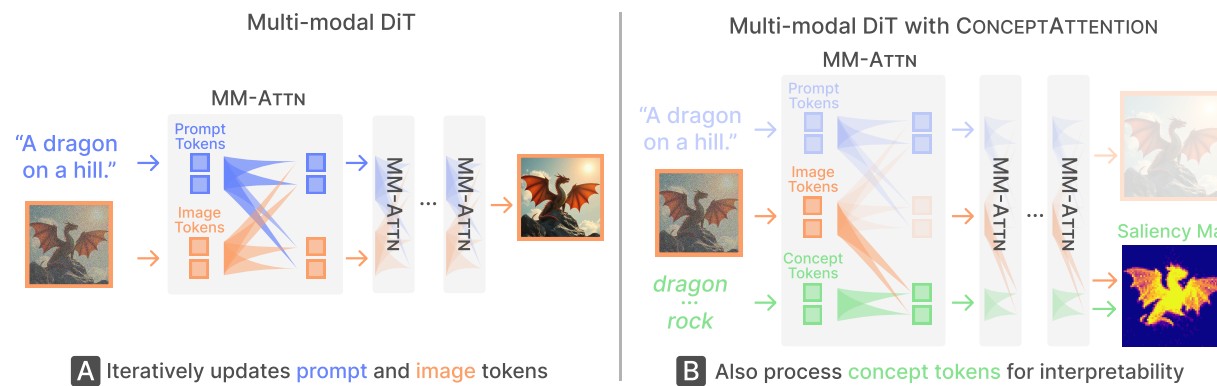

*Figure 2.* CONCEPTATTENTION **augments multi-modal DiTs with a sequence of concept embeddings that can be used to produce saliency maps.** (Left) An unmodified multi-modal attention (MMATTN) layer processes both **prompt** and **image** tokens. (Right) CONCEPTATTENTION augments these layers without impacting the image appearance to create a set of contextualized **concept** tokens.

of more recent multi-modal diffusion transformers (DiTs) remains underexplored. DiT-based models have recently replaced UNets (Ronneberger et al., 2015) as the state-of-the-art architecture for image generation, with models such as Flux (Labs, 2023) and SD3 (Esser et al., 2024) achieving breakthroughs in text-to-image generation. The rapid advancement and enhanced capabilities of DiT-based models highlight the critical importance of methods that improve their interpretability, transparency, and safety.

In this work, we propose CONCEPTATTENTION, a novel method that leverages the representations of multi-modal DiTs to produce high-fidelity saliency maps that localize textual concepts within images. Our method provides insight into the rich semantics of DiT representations. CONCEPTATTENTION is lightweight and requires no additional training, instead it repurposes the existing parameters of DiT attention layers. CONCEPTATTENTION works by producing a set of rich contextualized text embeddings each corresponding to visual concepts (e.g. "dragon", "sun"). By linearly projecting these *concept embeddings* and the image we can produce rich saliency maps that are even higher quality than commonly used cross attention maps.

We evaluate the efficacy of CONCEPTATTENTION in a zero-shot semantic segmentation task on real world images. We compare our interpretative maps against annotated segmentations to measure the accuracy and relevance of the attributions generated by our method. Our experiments and extensive comparisons demonstrate that CONCEPTATTENTION provides valuable insights into the inner workings of these otherwise complex black-box models. By explaining the meaning of the representations of generative models our method paves the way for advancements in interpretability, controllability, and trust in generative AI systems.

In summary, we contribute:

- **CONCEPTATTENTION, a method for interpreting**

**text-to-image diffusion transformers.** Our method requires no additional training, by leveraging the representations of multi-modal DiTs to generate highly interpretable saliency maps that depict the presence of arbitrary textual concepts (e.g. "dragon", "sky", etc.) in images (as shown in Figure 1).

- **The novel discovery that the output vectors of attention operations produce higher-quality saliency maps than cross attentions.** CONCEPTATTENTION repurposes the parameters of DiT attention layers to produce rich textual embeddings corresponding to different concepts, something that is uniquely enabled by multi-modal DiT architectures. By performing linear projections between these *concept embeddings* and image patch representations in the attention output space we can produce high quality saliency maps.

- **CONCEPTATTENTION achieves state-of-the-art performance in zero-shot segmentation on benchmarks like ImageNet Segmentation and Pascal VOC across multiple DiT architectures**. We achieve superior performance to a diverse set of zero-shot interpretability methods based on various foundation models like CLIP, DINO, and UNet-based diffusion models; this highlights the potential for the representations of DiTs to transfer to important downstream vision tasks like segmentation. We replicate our results quantitatively on both Flux and Stable Diffusion 3.5 Turbo.

- **CONCEPTATTENTION works with a video generation DiT model.** Additionally, we demonstrate qualitatively that CONCEPTATTENTION seamlessly generalizes to the CogVideoX (Yang et al., 2025) video generation MMDiT model, producing higher-quality saliency maps than native cross attention maps.

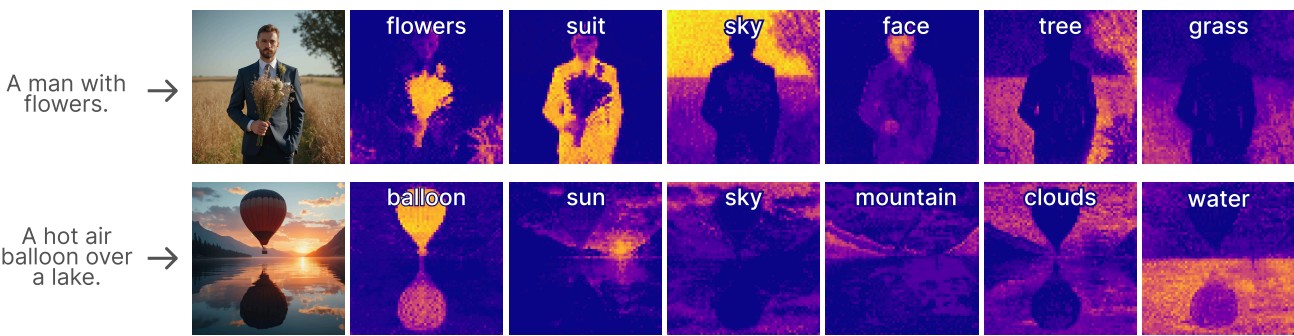

Figure 3. **CONCEPTATTENTION can generate high-quality saliency maps for multiple concepts simultaneously.** Additionally, our approach is not restricted to concepts in the prompt vocabulary.

## 2. Related Work

**Diffusion Model Interpretability** A fair amount of existing work attempts to interpret diffusion models. Some works investigate diffusion models from an analytic lens (Kadkhodaie et al., 2024; Wang et al., 2024), attempting to understand how diffusion models geometrically model the manifold of data. Other works attempt to understand how models memorize images (Carlini et al., 2023). An increasing body of work attempts to repurpose the representations of diffusion models for various tasks like classification (Li et al., 2023a), segmentation (Karazija et al., 2024), and even robotic control (Gupta et al., 2024). However, most relevant to our work is the substantial body of methods investigating how the representations of the neural network architectures underpinning diffusion can be used to garner insight into how these models work, steer their behavior, and improve their safety.

Numerous papers have observed that the cross attention mechanisms of UNet-based diffusion models like Stable Diffusion (Rombach et al., 2022) and SDXL (Podell et al., 2023) can produce interpretable saliency maps of textual concepts (Tang et al., 2022). Cross attention maps are used in a variety of image editing tasks like producing masks that localize objects of interest to edit (Dalva et al., 2024), controlling the layout of images (Chen et al., 2023; Epstein et al., 2023), altering the appearance of an image but retaining its layout (Hertz et al., 2022), and even generating synthetic data to train instruction based editing models (Brooks et al., 2023). Other works observe that performing interventions on cross attention maps can improve the faithfulness of images to prompts by ensuring attributes are assigned to the correct objects (Meral et al., 2024; Chefer et al., 2023). Additionally, it has been observed that self-attention layers of diffusion models encode useful information about the layout of images (Liu et al., 2024).

**Zero-shot Image Segmentation** In this work, we evaluate CONCEPTATTENTION on the task of zero-shot image segmentation, which is a natural way to assess the accuracy

of our saliency maps and the transferability of the representations of multi-modal DiT architectures to downstream vision tasks. This task also provides a good setting to compare to a variety of other interpretability methods for various foundation model architectures like CLIP (Radford et al., 2021), DINO (Caron et al., 2021), and diffusion models.

A variety of works train models from scratch for the task of image segmentation (Amit et al., 2022; Karazija et al., 2024) or attempt to fine-tune pretrained models (Baranchuk et al., 2022). Another line of work leverages diffusion models to generate synthetic data that can be used to train segmentation models that transfer zero-shot to new classes (Li et al., 2023b). While effective, these methods are training-based and thus do not provide as much insight into the representations of existing text-to-image generation models, which is the key motivation behind CONCEPTATTENTION.

A significant body of work attempts to improve the interpretability of CLIP vision transformers (ViTs) (Dosovitskiy et al., 2021). The authors of (Chefer et al., 2021) develop a method for generating saliency maps for ViT models, and they introduce an evaluation protocol for assessing the effectiveness of these saliency maps. This evaluation protocol centers around the ImageNet-Segmentation dataset (Guillaumin et al., 2014), and we extend this evaluation to the PascalVOC dataset (Everingham et al., 2015). They compare to a variety of zero-shot interpretability methods like GradCAM (Selvaraju et al., 2019), Layerwise-Relevance Propagation (Binder et al., 2016), raw attentions, and the Rollout method (Abnar & Zuidema, 2020). The authors of (Gandelsman et al., 2024) demonstrate an approach to expressing image patches in terms of textual concepts. We also compare our approach to zero-shot diffusion based methods (Tang et al., 2022; Wang et al., 2024) and the self-attention maps of DINO ViT models (Caron et al., 2021).

Another line of work performs unsupervised segmentation without any class or text conditioning by performing clustering of the embeddings of models (Cho et al., 2021; Hamilton et al., 2022; Tian et al., 2024). Despite not producing class predictions, these models are often evaluated on semantic

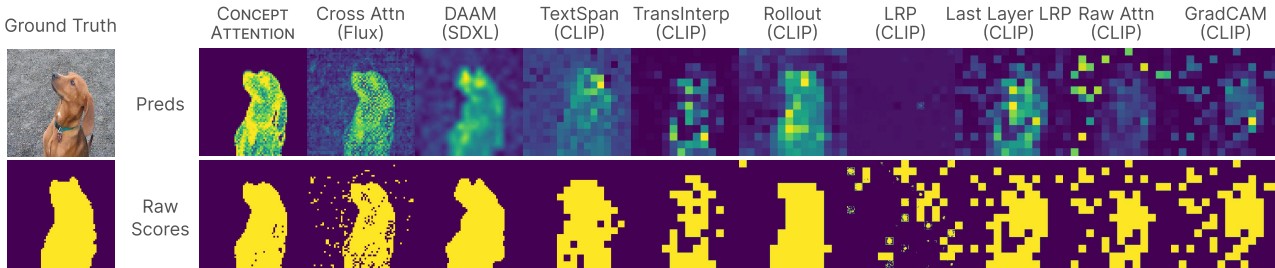

*Figure 4.* CONCEPTATTENTION **produces higher fidelity raw scores and saliency maps than alternative methods**, sometimes surpassing in quality even the ground truth saliency map provided by the ImageNet-Segmentation task. Top row shows the soft predictions of each method and the bottom shows the binarized predictions.

segmentation datasets by using approaches like Hungarian matching (Kuhn, 1955) to pair unlabeled segmentation predictions with the best matching ones in a multi-class semantic segmentation dataset. In contrast, CONCEPTATTENTION enables text conditioning so we do not compare to this family of methods. We also don't compare to models like SAM (Kirillov et al., 2023; Ravi et al., 2024) as it is trained on a large scale dataset.

## 3. Preliminaries

### 3.1. Rectified-Flow Models for Image Generation

Flux and Stable Diffusion 3 leverage multi-modal DiTs that are trained to parameterize rectified flow models. Throughout this paper we may refer to rectified flow models as diffusion models for convenience. These models attempt to generate realistic images from noise that correspond to given text prompts. Flow based models (Lipman et al., 2023) attempt to map a sample $x_1$ from a noise distribution $p_1$, typically $p_1 \sim \mathcal{N}(0, I)$, to a sample $x_0$ in the data distribution. Rectified flows (Liu et al., 2022) attempt to learn ODEs that follow straight paths between the $p_0$ and $p_1$, i.e.

$$z_t = (1-t)x_0 + t\epsilon, \epsilon \sim \mathcal{N}(0,1). \tag{1}$$

Flux and SD3 are trained using a conditional flow matching objective which can be expressed conveniently as

$$-\frac{1}{2}\mathbb{E}_{t\sim\mathcal{U}(t),\epsilon\sim\mathcal{N}(0,I)}[w_t\lambda'_t||\epsilon_\Theta(z_t,t) - \epsilon||^2] \tag{2}$$

where $\lambda'_t$ corresponds to a signal-to-noise ratio and $w_t$ is a time dependent-weighting factor. Above $\epsilon_\Theta(z_t, t)$ is parameterized by a multi-modal diffusion transformer network. The architecture of this model and it's properties is of primary interest in this work.

### 3.2. The Anatomy of a Multi-modal DiT Layer

Multi-modal DiTs like Flux and Stable Diffusion 3 leverage *multi-modal attention layers (*MMATTN*)* that process a combination of textual tokens and image patches. There

are two key classes of layers: one that keeps separate residual streams for each modality and one that uses a single stream. In this work, we take advantage of the properties of these dual stream layers, which we refer to as multi-modal attention layers (MMATTNs).

The input to a given layer is a sequence of image patch representations $x \in \mathbb{R}^{h \times w \times d}$ and prompt token embeddings $p \in \mathbb{R}^{l \times d}$. The initial prompt embeddings at the beginning of the network are formed by taking the T5 (Raffel et al., 2023) embeddings of the prompt tokens.

Following (Peebles & Xie, 2023), each MMATTN layer leverages a set of adaptive layer norm *modulation layers* (Xu et al., 2019), conditioned on the time-step and global CLIP vector. An adaptive layernorm operation is applied to the input image and text embeddings. The final modulated outputs are then residually added back to the original input. Notably, the image and text modalities are kept in separate residual streams. The exact details of this operation are omitted for brevity.

The key workhorse in MMATTN layers is the familiar multi-head self attention operation. The prompt and image embeddings have separate learned key, value, and query projection matrices which we refer to as $K_x, Q_x, V_x$ for images and $K_p, Q_p, V_p$ for text. The keys, queries, and values for both modalities are collectively denoted $q_{xp}$, $k_{xp}$, and $v_{xp}$, where for example $k_{xp} = [K_x x_1, \ldots, K_p p_1 \ldots]$. A self attention operation is then performed

$$o_x, o_p = \text{softmax}(q_{xp}k_{xp}^T)v_{xp} \tag{3}$$

Here we refer to $o_x$ and $o_p$ as the *attention output* vectors. Another linear layer is then applied to these outputs and added to a separate residual streams weighted according to the output of the modulation layer. This gives us updated embeddings $x^{L+1}$ and $p^{L+1}$ which are given as input to the next layer.

## 4. Methods

We introduce CONCEPTATTENTION, a method for generating high quality saliency maps depicting the location of

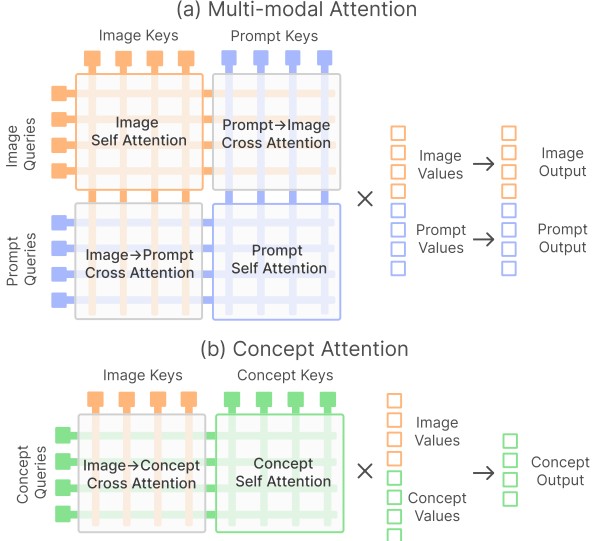

*Figure 5.* (a) MMATTN combines cross and self attention operations between the prompt and image tokens. (b) Our CONCEPTATTENTION allows the concept tokens to incorporate information from other concept tokens and the image tokens, but not the other way around.

textual concepts in images. CONCEPTATTENTION works by creating a set of contextualized *concept embeddings* for simple textual concepts (e.g. "cat", "sky", etc.). These concept embeddings are sequentially updated alongside the text and image embeddings, so they match the structure that each MMATTN layer expects. However, unlike the text prompt, concept embeddings do not impact the appearance of the image. We can produce high-fidelity saliency maps by projecting image patch representations onto the concept embeddings. CONCEPTATTENTION requires no additional training and has minimal impact on model latency and memory footprint. A high level depiction of our methodology is shown in Figure 2.

### 4.1. Using CONCEPTATTENTION

The user specifies a set of $r$ single token concepts, like "cat", "sky", etc. which are passed through a T5 encoder to produce an initial embedding $c^0$ for each concept. For each MMATTN layer (indexed by $L$) we layer-normalize the input concept embeddings $c^L$ and repurpose the text prompt's projection matrices (i.e. $K_p, Q_p, V_p$), to produce a set of keys, values, and queries

$$k_c = [K_p c_1, \dots], q_c = [Q_p c_1, \dots], v_c = [V_p c_1, \dots] \in \mathbb{R}^{r \times d}. \quad (4)$$

**One-directional Attention Operation** We would like to update our concept embeddings so they are compatible with subsequent layers, but also prevent them from impacting the image tokens. Let $k_x$ and $v_x$ be the keys and values of

the image patches $x$ respectively. We can concatenate the image and concept keys to get

$$k_{xc} = [K_x x_1 \dots, K_x x_n, K_p c_1 \dots, K_p c_r] \quad (5)$$

and the image and concept values to get

$$v_{xc} = [V_x x_1 \dots, V_x x_n, V_p c_1 \dots, V_p c_r] \quad (6)$$

We can then perform the following attention operation

$$o_c = \text{softmax}(q_c k_{xc}^T) v_{xc} \quad (7)$$

which produces a set of *concept output embeddings*.

Notice, that instead of just performing a cross attention (i.e. $\text{softmax}(q_c k_x^T) v_x$) our approach leverages both cross attention from the image patches to the concepts and self attention among the concepts. We found that performing both operations improves performance on downstream evaluation tasks like segmentation (See Table 4). We hypothesize this is because it allows the concept embeddings to repel from each other, avoiding redundancy between concepts.

Meanwhile, the image patch and prompt tokens ignore the concept tokens and attend only to each other as in

$$o_x, o_p = \text{softmax}(q_{xp} k_{xp}^T) v_{xp}. \quad (8)$$

A diagram of these operations is shown in Figure 5(b).

**A Concept Residual Stream** The above operations create a residual stream of concept embeddings distinct from the image and patch embeddings. Following the pretrained transformer's design, after the MMATTN we apply another projection matrix $P$ and MLP, adding the result residually to $c^L$. We apply an adaptive layernorm at the end of the attention operation which outputs several values: a scale $\gamma$, shift $\beta$, and some gating values $\alpha_1$ and $\alpha_2$. The residual stream is then updated as

$$c^{L+1} \leftarrow c^L + \alpha_1(P o_c) \quad (9)$$

$$c^{L+1} \leftarrow c^{L+1} + \alpha_2 \text{MLP}\left((1 + \gamma) \text{lnorm}(c^{L+1}) + \beta\right) \quad (10)$$

where $\leftarrow$ denotes assignment. The parameters from each of our modulation, projection, and MLP layers are the same as those used to process the text prompt.

**Saliency Maps in the Attention Output Space** These concept embeddings can be combined with the image patch embeddings to produce saliency maps for each layer $L$. Specifically, we found that taking a simple dot-product similarity between the image output vectors $o_x$ and concept output vectors $o_c$ produces high-quality saliency maps

| Method | Architecture | ImageNet-Segmentation | | | PascalVOC (Single Class) | | |
|---|---|---|---|---|---|---|---|
| | | Acc ↑ | mIoU↑ | mAP↑ | Acc ↑ | mIoU↑ | mAP↑ |
| LRP (Binder et al., 2016) | CLIP ViT | 51.09 | 32.89 | 55.68 | 48.77 | 31.44 | 52.89 |
| Partial-LRP (Binder et al., 2016) | CLIP ViT | 76.31 | 57.94 | 84.67 | 71.52 | 51.39 | 84.86 |
| Rollout (Abnar & Zuidema, 2020) | CLIP ViT | 73.54 | 55.42 | 84.76 | 69.81 | 51.26 | 85.34 |
| ViT Attention (Dosovitskiy et al., 2021) | CLIP ViT | 67.84 | 46.37 | 80.24 | 68.51 | 44.81 | 83.63 |
| GradCAM (Selvaraju et al., 2020) | CLIP ViT | 64.44 | 40.82 | 71.60 | 70.44 | 44.90 | 76.80 |
| TextSpan (Gandelsman et al., 2024) | CLIP ViT | 75.21 | 54.50 | 81.61 | 75.00 | 56.24 | 84.79 |
| TransInterp (Chefer et al., 2021) | CLIP ViT | 79.70 | 61.95 | 86.03 | 76.90 | 57.08 | 86.74 |
| CLIPasRNN (Sun et al., 2024) | CLIP ViT | 74.05 | 58.80 | 84.80 | 61.76 | 41.48 | 76.57 |
| OVAM (Marcos-Manchón et al., 2024) | SDXL UNet | 79.41 | 65.02 | 88.12 | 73.50 | 58.12 | 87.91 |
| DINO SA (Caron et al., 2021) | DINO ViT | 81.97 | 69.44 | 86.12 | 80.71 | 64.33 | 88.90 |
| DINOv2 SA (Oquab et al., 2024) | DINOv2 ViT | 77.39 | 63.12 | 84.19 | 79.65 | 57.61 | 87.26 |
| DINOv2 Reg SA (Darcet et al., 2024) | DINOv2 Reg | 72.04 | 56.31 | 80.83 | 77.16 | 56.60 | 86.35 |
| iBOT SA (Zhou et al., 2022) | iBOT ViT | 76.34 | 61.73 | 82.04 | 74.96 | 55.80 | 85.26 |
| DAAM (Tang et al., 2022) | SDXL UNet | 78.47 | 64.56 | 88.79 | 72.76 | 55.95 | 88.34 |
| DAAM (Tang et al., 2022) | SD2 UNet | 64.52 | 47.62 | 78.01 | 64.28 | 45.01 | 83.04 |
| Cross Attention | Flux DiT | 74.92 | 59.90 | 87.23 | 80.37 | 54.77 | 89.08 |
| Cross Attention | SD3.5 DiT | 77.80 | 63.67 | 83.50 | 80.22 | 61.46 | 86.97 |
| CONCEPTATTENTION | SD3.5 DiT | 81.92 | 67.47 | **90.79** | 83.90 | 69.93 | 90.02 |
| CONCEPTATTENTION | Flux DiT | **83.07** | **71.04** | 90.45 | **87.85** | **76.45** | **90.19** |

*Table 1.* CONCEPTATTENTION outperforms a variety of Diffusion, DINO, and CLIP ViT interpretability methods on ImageNet-Segmentation and PascalVOC (Single Class).

$$\phi(o_x, o_c) = \text{softmax}(o_x o_c^T). \quad (11)$$

This is in contrast to cross attention maps which are between the image keys $k_x$ and prompt queries $q_p$.

We can aggregate the information from multiple layers by averaging them $\frac{1}{|L|} \sum_{L=1}^{|L|} \phi(o_x^L, o_c^L)$ where $|L|$ denotes the number of MMATTN layers (Flux has $|L| = 18$). These attention output space maps are unique to MM-DiT models as they leverage *concept embeddings* corresponding to textual concepts which fundamentally can not be produced by UNet-based models.

### 4.2. Limitations of Raw Cross Attention Maps

For multi-modal DiT architectures, we could additionally consider using the raw cross attention maps

$$\phi(k_x, q_p) = \text{softmax}(q_p k_x^T) \quad (12)$$

to produce saliency maps. However, these have a key limitation in that their vocabulary is limited to the tokens in the user's prompt. Unlike UNet-based models, multi-modal DiTs sequentially update a set of prompt embeddings with each MMATTN layer. This makes it difficult to produce cross attention maps for an open-set of concepts, as you would need to add the concept to the prompt sequence which would then change the appearance of the image. CONCEPTATTENTION overcomes this key limitation, and makes the additional discovery that the output space of attention mechanisms produces high-fidelity saliency maps.

| Method | Acc↑ | mIoU↑ |
|---|---|---|
| TextSpan | 73.84 | 38.10 |
| DAAM | 62.89 | 10.97 |
| Flux Cross Attention | 79.52 | 27.04 |
| CONCEPTATTENTION | **86.99** | **51.39** |

*Table 2.* **CONCEPTATTENTION outperforms alternative methods on images with multiple classes from PascalVOC.** Notably, the margin between CONCEPTATTENTION and other methods is even higher for this task than when a single class is in each image.

## 5. Experiments

### 5.1. Implementation Details

**Flux DiT** For most of our experiments we use the Flux DiT architecture implemented in PyTorch (Paszke et al., 2019). In particular, we use the distilled Flux-Schnell model. We encode real images with the DiT by first mapping them to the VAE latent space and then adding varying degrees of Gaussian noise before passing them through the Flux DiT. We then cache all of the concept output $o_c$ and image output vectors $o_x$ from each MMATTN layer. At the end of generation we then construct our concept saliency maps for each layer and average them over all layers of interest. In our experiments we leverage the activations from the last 10 of the 18 MMATTN layers.

**Stable Diffusion 3.5 Turbo** We found that our approach replicated on the Stable Diffusion 3.5 Turbo (Esser et al., 2024) DiT architecture (Figure 6).

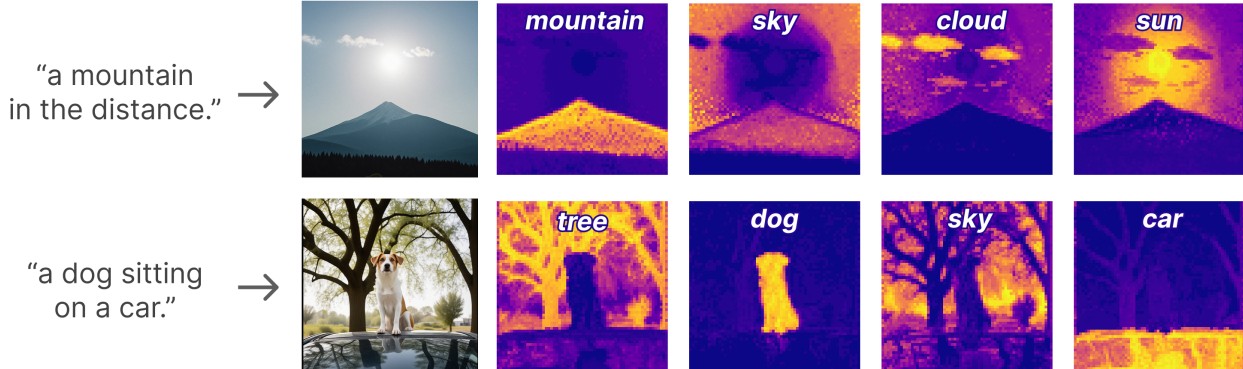

*Figure 6.* **CONCEPTATTENTION is capable of generating high quality saliency maps with Stable Diffusion 3.5 Turbo.** Furthermore, the top example highlights a potential failure case of CONCEPTATTENTION. The concepts "sky", "mountain", and "sun" all semantically overlap, resulting in unclear object boundaries.

| Space | Softmax | Acc↑ | mIoU↑ | mAP↑ |
|---|---|---|---|---|
| CA | | 66.59 | 49.91 | 73.17 |
| CA | ✓ | 74.92 | 59.90 | 87.23 |
| Value | | 45.93 | 29.81 | 65.79 |
| Value | ✓ | 45.78 | 29.68 | 39.61 |
| Output | | 78.75 | 64.95 | 88.39 |
| Output | ✓ | **83.07** | **71.04** | **90.45** |

*Table 3.* **The output space of DiT attention layers produces more transferable representations than cross attentions.** We explore the transferability of several representation spaces of a DiT: the cross attentions (CA), the value space, and the output space. We performed linear projections of the image patches and concept vectors in each of these spaces and evaluated their performance on ImageNet-Segmentation.

**CogVideo X** CONCEPTATTENTION generalizes to the CogVideoX (Yang et al., 2025) multi-modal DiT video generation model. The only change we make is additionally averaging over the added frame dimension.

## 5.2. Zero-shot Image Segmentation

We are interested in investigating (1) the efficacy of CONCEPTATTENTION to generate highly localized and semantically meaningful saliency maps, and (2) understand the transferability of multi-modal DiT representations to important downstream vision tasks. Zero-shot image segmentation is a natural choice for achieving these goals.

**Datasets** We leverage two key datasets zero-shot image segmentation datasets. First, we use a commonly used (Gandelsman et al., 2024; Chefer et al., 2021) zero-shot segmentation benchmark called ImageNet-Segmentation (Guillaumin et al., 2014). It is composed of 4,276 images from 445 categories. Each image primarily depicts a single central object or concept, which makes it a good method for comparing CONCEPTATTENTION to a variety of methods which

| CA | SA | Acc↑ | mIoU↑ | mAP↑ |
|---|---|---|---|---|
| | | 52.63 | 35.72 | 70.21 |
| | ✓ | 51.68 | 34.85 | 69.36 |
| ✓ | | 76.51 | 61.96 | 86.73 |
| ✓ | ✓ | **83.07** | **71.04** | **90.45** |

*Table 4.* **CONCEPTATTENTION performs best when we utilize both cross and self attention.** We tested the effectiveness of performing just a cross attention operation between the concepts and image tokens, just a self attention among the concepts, both cross and self attention, and neither. We found that doing both operations leads to the best results. Metrics are computed on the ImageNet Segmentation benchmark.

generate a single saliency map that are unable to generate class-specific segmentation maps. For the second dataset we leverage PascalVOC 2012 benchmark (Everingham et al., 2015). We investigate both a single class (930 images) and multi-class split (1,449 images) of this dataset. Many methods (e.g. DINO) do not condition their saliency map on class, so for these methods we restrict our evaluation to examples only containing a single class and the background. For methods that can accept text as conditioning we evaluate on the full dataset.

**Key Baseline Methods** We compare our approach to a variety of zero-shot interpretability methods which leverage several different multi-modal foundation models. We compare to numerous interpretability methods compatible with CLIP: Layerwise Relevance Propagation (LRP) (Binder et al., 2016), LRP on just the final-layer of a ViT (Partial-LRP), Attention Rollout (Rollout) (Abnar & Zuidema, 2020), Raw Vision Transformer Attention (ViT Attention) (Dosovitskiy et al., 2021), GradCAM (Selvaraju et al., 2019), TextSpan (Gandelsman et al., 2024), CLIP as RNN (Sun et al., 2024), and the Transformer Attribution method from (Chefer et al., 2021) (TransInterp). We also

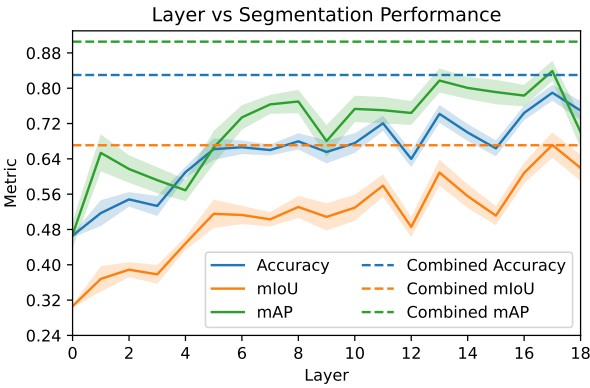 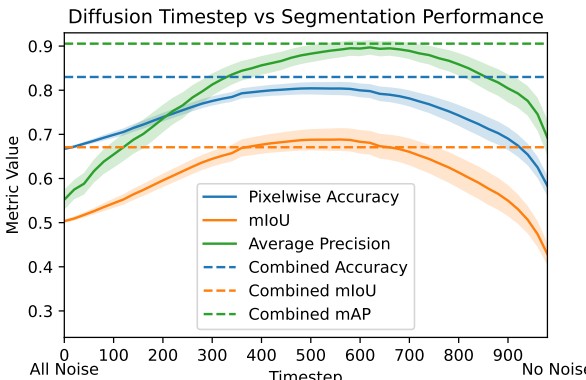

*Figure 7.* **(Left) Later MMATTN layers encode richer features for zero-shot segmentation.** We investigated the impact of using features from various MMATTN layers and found that deeper layers lead to better performance on segmentation metrics like pixelwise accuracy, mIoU, and mAP. We also found that combining the information from all layers further improves performance. **(Right) Optimal segmentation performance requires some noise to be present in the image.** We evaluated the performance of CONCEPTATTENTION by encoding samples from a variety of timesteps (determines the amount of noise). Interestingly, we found that the optimal amount of noise was not zero, but in the middle to later stages of the noise schedule.

compare to UNet-based interpretability methods that aggregates information from UNet cross attention layers called DAAM (Tang et al., 2022) on both SDXL (Podell et al., 2023) and SD2, and OVAM (Li et al., 2023b) with SDXL. We compare to the self-attention maps of various DINO models: DINOv1 (Caron et al., 2021), DINOv2 (Oquab et al., 2024), and DINOv2 with registers (Darcet et al., 2024). Finally, we compare to the raw cross attention maps produced by Flux and Stable Diffusion 3.5 Turbo.

**Single Object Image Segmentation**  For our first task we closely follow the established evaluation framework from (Gandelsman et al., 2024) and (Chefer et al., 2021). We perform this evaluation setup on both ImageNet-Segmentation and a subset of 930 PascalVOC images containing only a single class. For each method we assume the class present in the image is known and use simplified descriptions of each ImageNet class (e.g. "Maltese dog" → "dog) this allows the concepts to be captured by a single token. We construct a concept vocabulary for each image composed of this target class and a set of fixed background concepts for all examples (e.g. "background", "grass", "sky").

**Quantitative Evaluation**  Each method produces a set of scalar *raw scores* for each image patch which we then threshold based on the mean value to produce a binary segmentation prediction. We compare each method using standard segmentation evaluation metrics, namely: mean Intersection over Union (mIoU), patch/pixelwise accuracy (Acc), and mean Average Precision (mAP). Accuracy alone is an insufficient metric as our dataset is highly imbalanced, mIoU is significantly better, and mAP captures threshold agnostic segmentation capability. We found that CONCEPTATTENTION significantly out performs all of the baselines we tested

across all three metrics (Table 1). This is true for diffusion, CLIP, and DINO based interpretability methods.

**Qualitative Evaluation**  We show qualitative results comparing the segmentation performance to each baseline in Figure 4 and more qualitative results in Appendix B. It is worth noting that the qualitative segmentation results highlight (a) the ambiguity of zero-shot image segmentation, and (b) the limitations of human data annotation. For example, Figure 4 shows our method does not segment the part of the dog between the ears and it's body, while the ground truth does.

**Multi Object Image Segmentation**  We also wanted to evaluate the capabilities of our method at differentiating between multiple classes in an image. However, only a subset of methods produce distinct saliency maps for open ended classes. For this experiment we compare to DAAM using a SDXL backbone, TextSpan using a CLIP backbone, and the raw cross attentions of Flux. Instead of binarizing the image to produce segmentations, for each patch we predict the concept with the highest score. We used pixelwise accuracy and mIoU as our evaluation metrics and found that our method significantly outperformed the baselines (Table 2). We also show qualitative results of our approach differentiating between multiple concepts in a single image in Figures 1, 3 and we show more results in Appendix B.

### 5.3. Ablation Studies

We perform several ablation studies to investigate the impact of various architectural choices and hyperparameters on the performance of CONCEPTATTENTION.

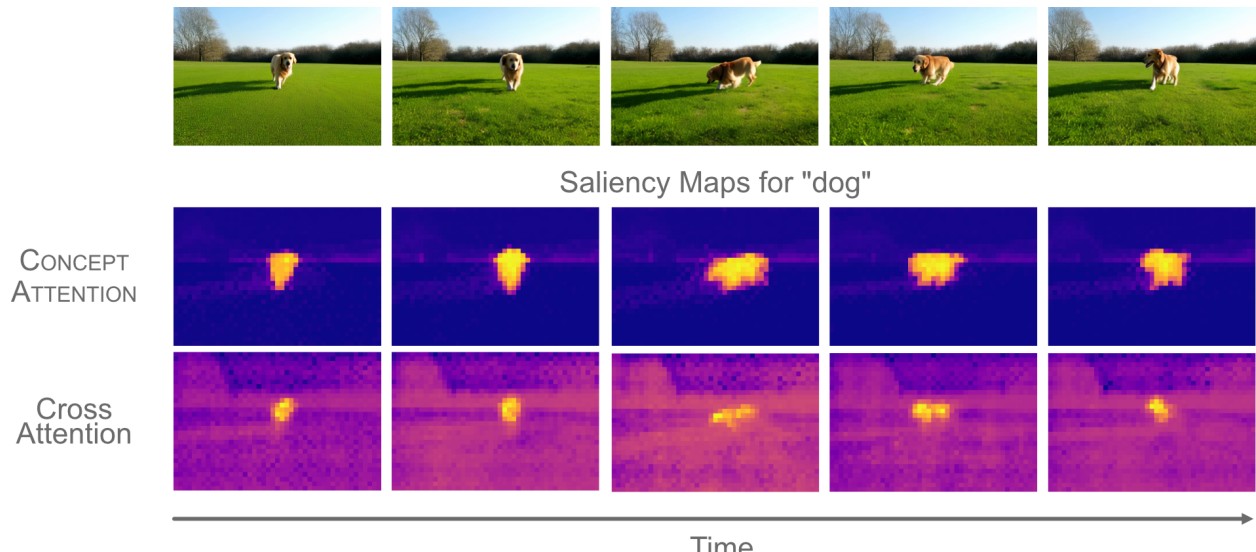

*Figure 8.* **CONCEPTATTENTION generalizes seamlessly to video generation MMDiT models like CogVideoX.** We apply CONCEPTATTENTION to a CogVideoX (Yang et al., 2025) video generation model. We take several frames from the video and compare the saliency maps generated by CONCEPTATTENTION to the model's internal cross attention maps.

**Impact of Layer Depth on Segmentation** We hypothesized that deeper MMATTN layers in the DiT would have more refined representations that better transfer to segmentation. This was confirmed by our evaluation (Figure 7). We pull features from each diffusion layer and evaluated the segmentation performance on ImageNet-Segmentation. We also compare the performance of combining all layers simultaneously, which we found performs better than any individual layer.

**Impact of Diffusion Timestep on Segmentation** We add Gaussian noise to encoded images before passing them to the DiTs, this conforms with the expectations of the models. Intuitively one might expect the later timesteps (less noise) to have much higher segmentation performance as less information about the original image is corrupted. However, we found that the middle diffusion timesteps best (Figure 7). Throughout the rest of our experiments we use timestep 500 out of 1000 following this result.

**Concept Attention Operation Ablations** We compared the performance on the ImageNet Segmentation benchmark of performing (a) just cross attention from the image patches to the concept vectors, (b) just self attention, (c) no attention operations, and (d) both cross and self attention. Our results seen in Table 4 indicate that using a combination of both cross and self attention achieves the best performance. We also investigated the impact of applying a pixelwise softmax operation over our predicted segmentation coefficients. We found that it slightly improves segmentation performance in the attention output space and significantly improves performance for the cross attention maps (Table 3)

### 5.4. Video Model Results

We include qualitative results demonstrating the efficacy of CONCEPTATTENTION on the CogVideoX video generation multi-modal DiT model (Figure 8). Also see Figures 16 and 17 in the Appendix.

### 5.5. Limitations

The primary limitation of CONCEPTATTENTION is that it struggles to differentiate between very similar textual concepts. For example, for a photo with a sky with the sun in it, the model does not necessarily know where the boundary of the sun resides, instead capturing a halo around the sun (Figure 6). Additionally, when no relevant concept is present, CONCEPTATTENTION will select the most similar one even if it is incorrect (Figure 15 in the Appendix).

## 6. Conclusion

We introduce CONCEPTATTENTION, a method for interpreting the rich features of multi-modal DiTs. Our approach allows a user to produce high quality saliency maps of an open-set of textual concepts that shed light on how a diffusion model "sees" an image. We perform an extensive evaluation of the saliency maps on zero-shot segmentation and find that they significantly outperform a variety of other zero-shot interpretability methods. Our results suggest the potential for DiT models to act as powerful and interpretable image encoders with representations that are transferable zero-shot to tasks like image segmentation.

## Impact Statement

Generative models for images have numerous ethical concerns: they have the potential to spread misinformation through realistic fake images (i.e. deepfakes), they may disrupt different creative industries, and have the potential to reinforce existing social biases present in their training data. Our work directly serves to improve the transparency of these models, and we believe our work could be used to understand the biases present in models.

## Acknowledgments

This paper is supported by the National Science Foundation Graduate Research Fellowship. This work was also supported in part by Cisco, NSF #2403297, gifts from Google, Amazon, Meta, NVIDIA, Avast, Fiddler Labs, Bosch.

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

# A. More In-depth Explanation of Concept Attention

We show pseudo-code depicting the difference between a vanilla multi-modal attention mechanism and a multi-modal attention mechanism with concept attention added to it.

(a) Multi-Modal Attention

(b) Multi-modal Attention with Concept Attention

```
def multi_modal_attn(img, txt):
    # Compute the keys, queries, and values
    img_k, img_q, img_v = img_projection(img)
    txt_k, txt_q, txt_v = txt_projection(txt)

    # Concat the image and text keys, queries, and vals
    img_txt_k = concat([img_k, txt_k])
    img_txt_q = concat([img_q, txt_q])
    img_txt_v = concat([img_v, txt_v])
    # Perform self attention on combined sequence
    attn_out = self_attention(img_txt_k, img_txt_q, img_txt_v)
    # Unpack the attention outputs
    img = attn_out[:img.shape[0]], attn_out[img.shape[0]:]

    return img, txt
```

```
+ def multi_modal_attn_with_concept_attn(img, txt, concepts):
      # Compute the keys, queries, and values
      img_k, img_q, img_v = img_projection(img)
      txt_k, txt_q, txt_v = txt_projection(txt)
+     concept_k, concept_q, concept_v = txt_projection(concepts)
      # Concat the image and text keys, queries, and vals
      img_txt_k = concat([img_k, txt_k])
      img_txt_q = concat([img_q, txt_q])
      img_txt_v = concat([img_v, txt_v])
      # Perform self attention on combined sequence
      attn_out = self_attention(img_txt_k, img_txt_q, img_txt_v)
      # Unpack the attention outputs
      img, txt = attn_out[:img.shape[0]], attn_out[img.shape[0]:]
+     # Concatenate the image and concept keys and values
+     img_concept_k = concat([img_k, concept_k])
+     img_concept_v = concat([img_v, concept_v])
+     # Perform the concept attention
+     concept_attn_map = matmul(concept_q, img_concept_k.T)
+     concept_attn_map = softmax(concept_attn_map, axis=-1) * scale
+     concepts = matmul(concept_attn_map, img_concept_v)
+
+     return img, txt, concepts
```

*Figure 9.* **Pseudo-code depicting the (a) multi-modal attention operation used by Flux DiTs and (b) our CONCEPTATTENTION operation.** We leverage the parameters of a multi-modal attention layer to construct a set of contextualized concept embeddings. The concepts query the image tokens (cross-attention) and other concept tokens (self-attention) in an attention operation. The updated concept embeddings are returned in addition to the image and text embeddings.

## B. More Qualitative Results

Here we show a variety of qualitative results for our method that we could not fit into the original paper.

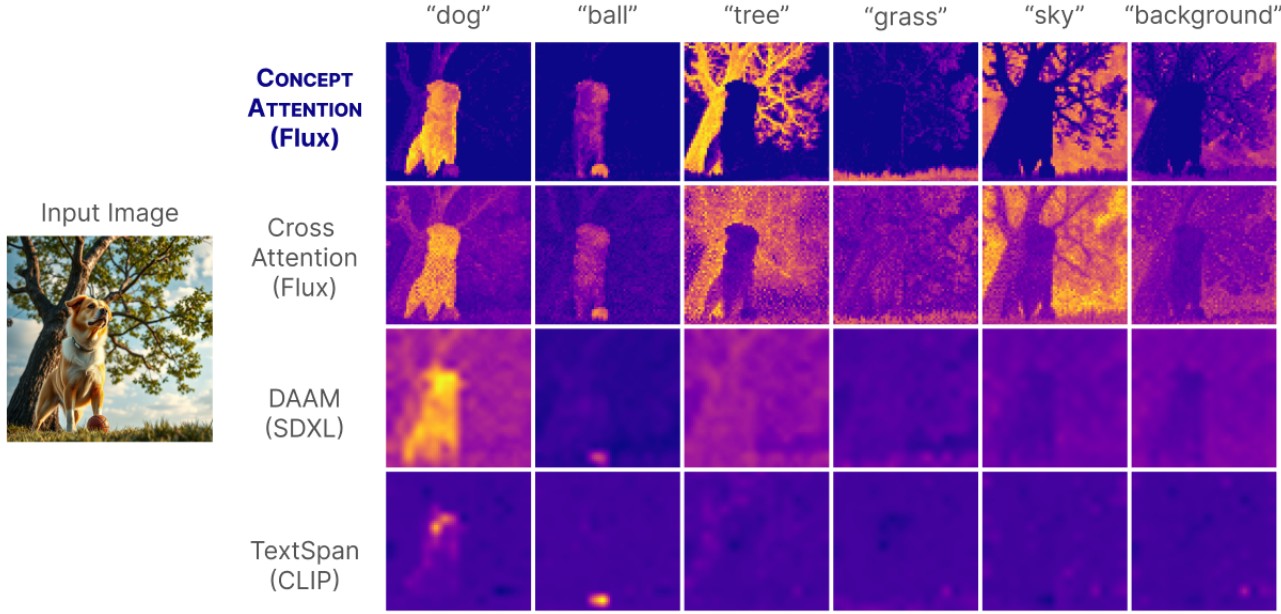

*Figure 10.* A qualitative comparison between our method and several others.

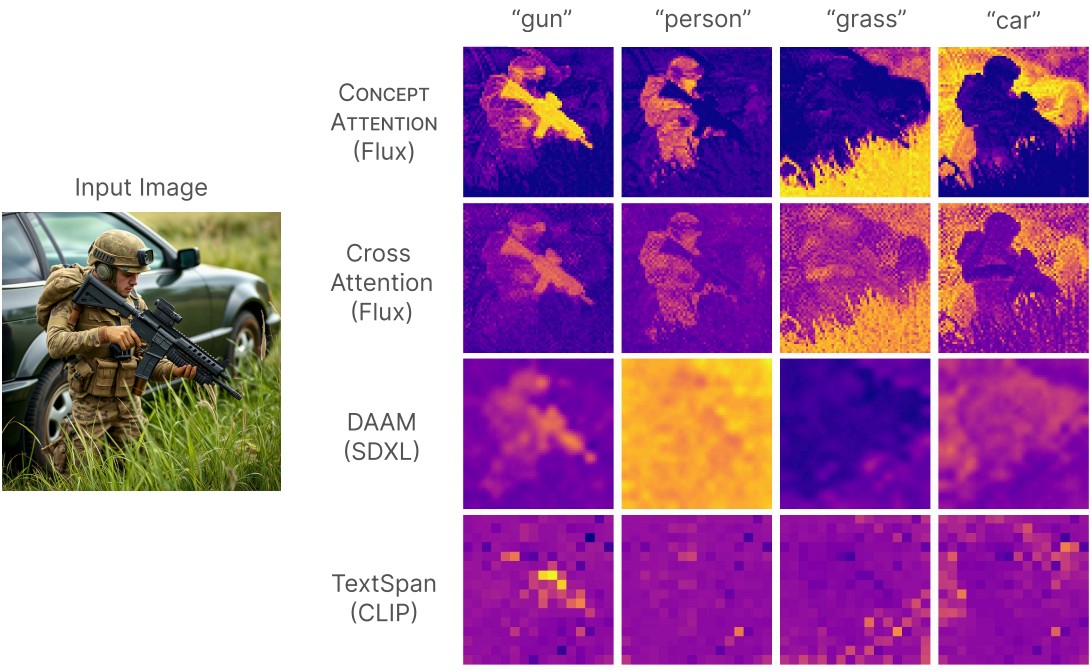

*Figure 11.* A qualitative comparison between our method and several others.

## C. Concept Attention on Video Generation Models

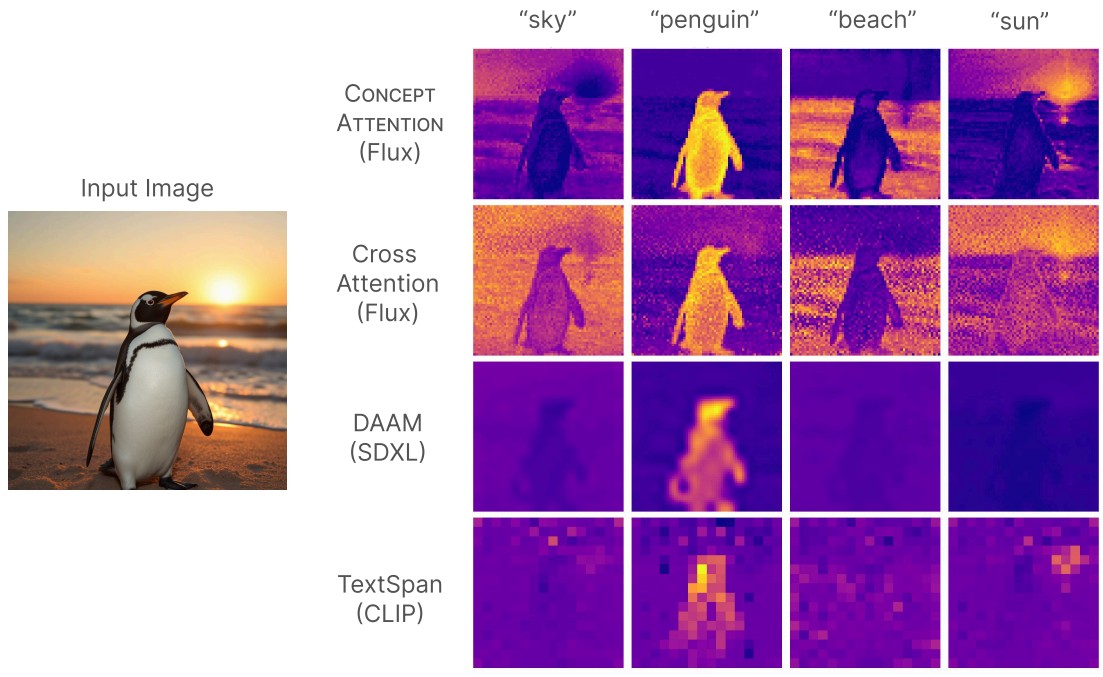

*Figure 12.* A qualitative comparison between our method and several others.

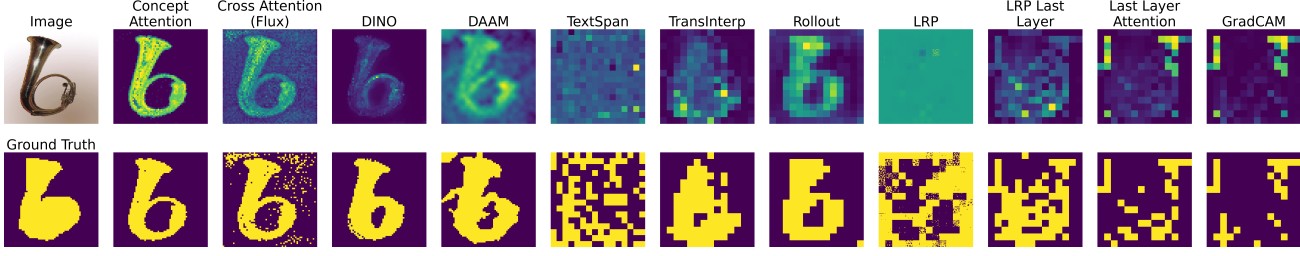

*Figure 13.* A qualitative comparison between numerous baselines on ImageNet Segmentation Images. The top row shows the soft predictions of each method and the bottom shows the binarized segmentation predictions.

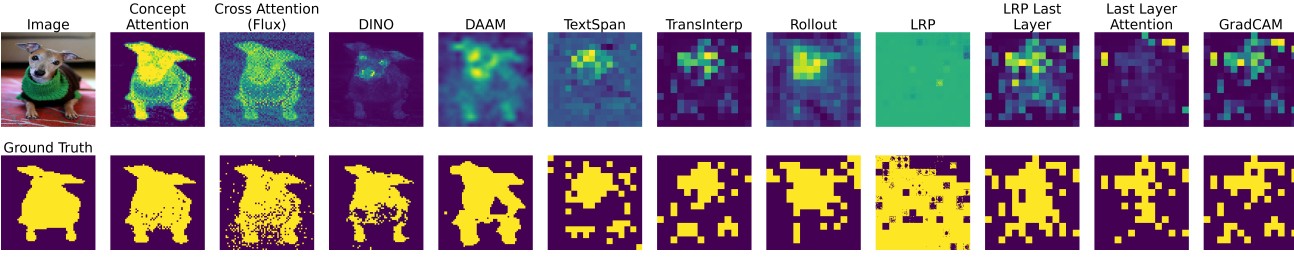

*Figure 14.* A qualitative comparison between numerous baselines on ImageNet Segmentation Images. The top row shows the soft predictions of each method and the bottom shows the binarized segmentation predictions.

Correct concept "bike" chosen over similar concept "car" when both are given

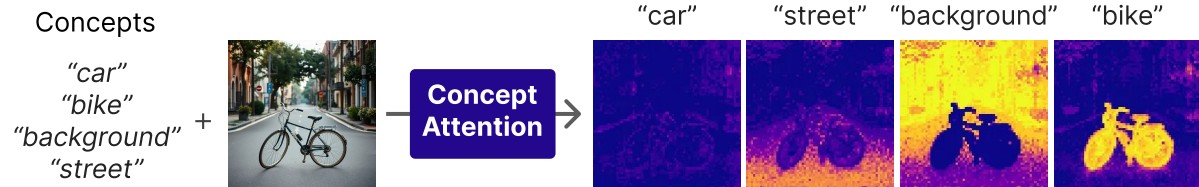

Closest concept "car" chosen when correct concept "bike" is not present

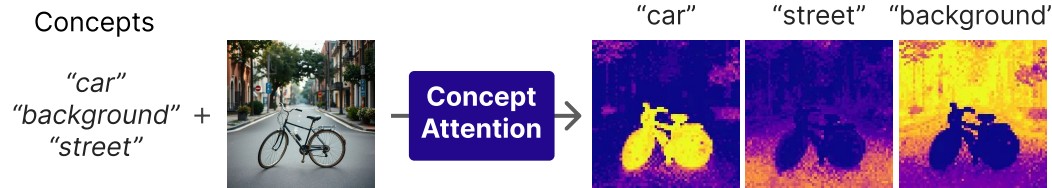

*Figure 15.* **The behavior of CONCEPTATTENTION when multiple relevant concepts are present and when no relevant one is.** When multiple similar concepts are given, like "car" and "bike", the most similar one will be chosen. However, when no relevant concept is presented, CONCEPTATTENTION will fall back on the most relevant one, in this case "car" for the bike patches.

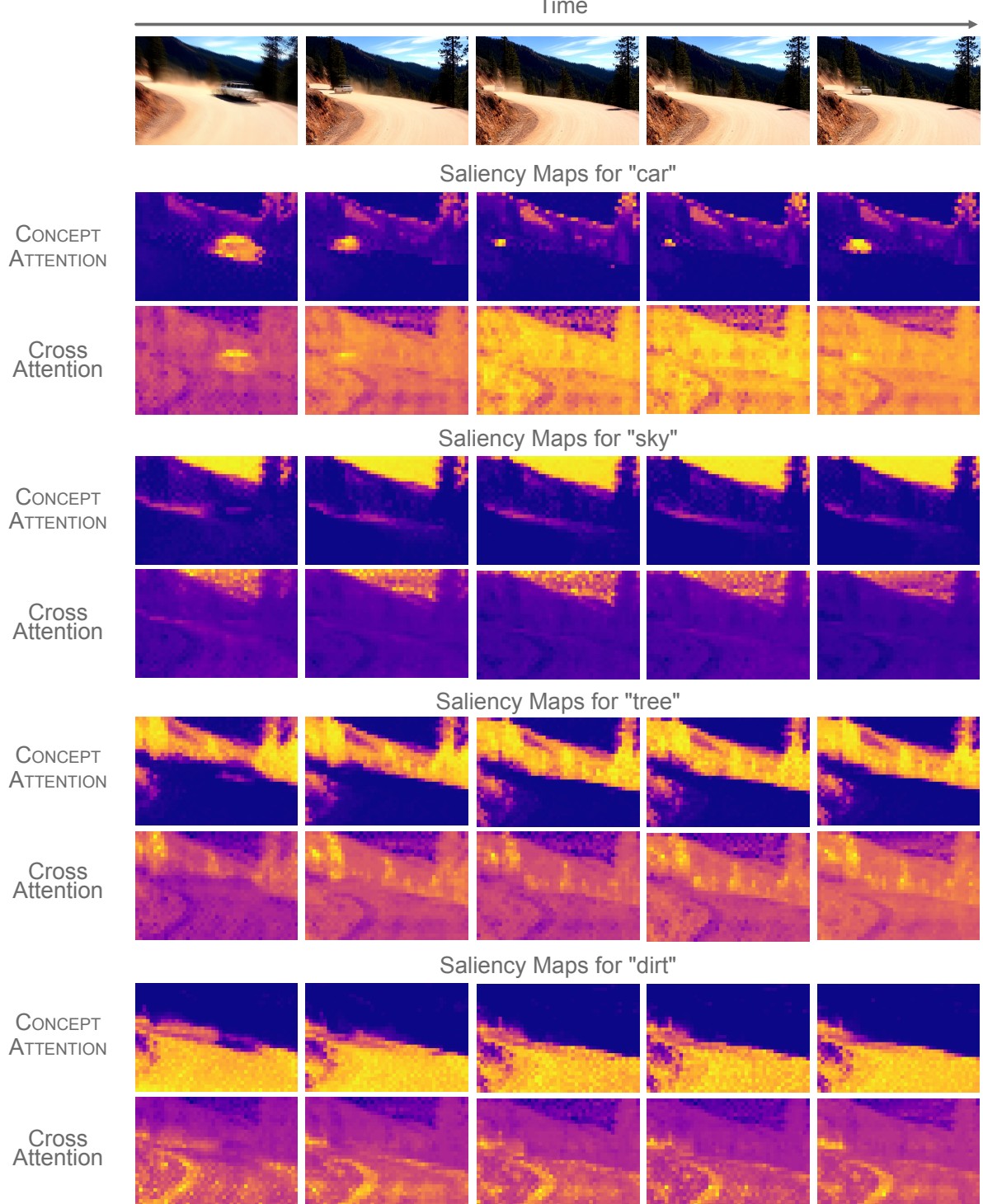

*Figure 16.* **CONCEPTATTENTION generalizes seamlessly to video generation MMDiT models like CogVideoX.** We apply CON-CEPTATTENTION to a CogVideoX (Yang et al., 2025) video generation model. We take several frames from the video and compare the saliency maps generated by CONCEPTATTENTION to the model's internal cross attention maps.

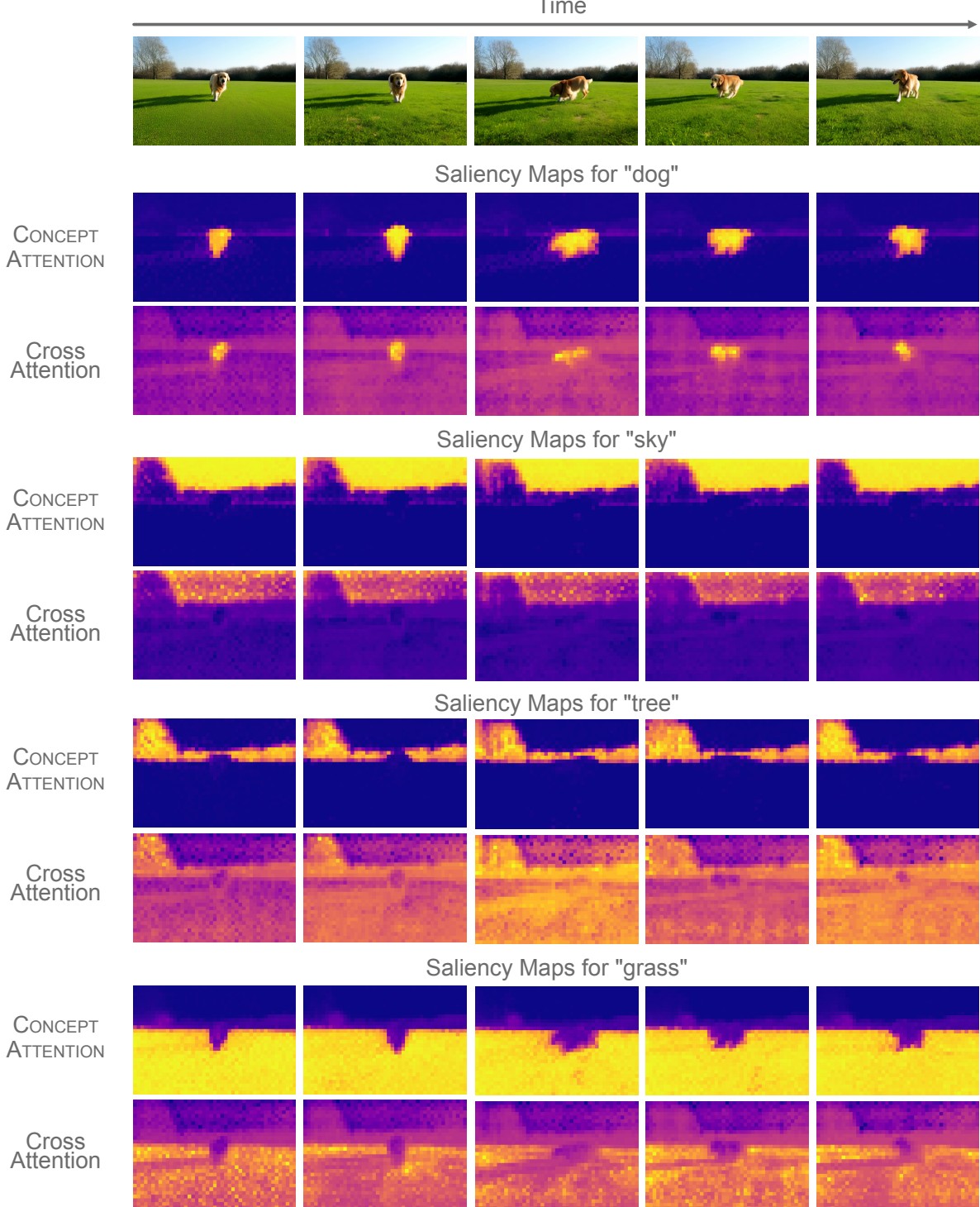

*Figure 17.* **CONCEPTATTENTION generalizes seamlessly to video generation MMDiT models like CogVideoX.** We apply CONCEPTATTENTION to a CogVideoX (Yang et al., 2025) video generation model. We take several frames from the video and compare the saliency maps generated by CONCEPTATTENTION to the model's internal cross attention maps.

