# OpenReview forum: "ConceptAttention: Diffusion Transformers Learn Highly Interpretable Features"
_ICML.cc/2025/Conference — ICML 2025 oral_

### Official Review · Reviewer_Njw6 · 2025-02-21

**Overall Recommendation:** 4

**Summary:**

The paper introduces a method that repurposes the attention mechanisms of multi-modal diffusion transformers (DiTs) to generate highly precise and interpretable saliency maps. Instead of relying solely on traditional cross attention, CONCEPTATTENTION leverages both cross and self attention in the output space of DiT layers to produce contextualized concept embeddings. These embeddings effectively map textual concepts (like “cat” or “sky”) onto corresponding regions in images. The method operates without additional training and is lightweight, making it a practical tool for enhancing the interpretability of diffusion models. Empirical results show that CONCEPTATTENTION achieves state-of-the-art performance in zero-shot image segmentation tasks on benchmarks such as ImageNet-Segmentation and PascalVOC, outperforming several existing approaches.

**Claims And Evidence:**

The experimental evidence largely backs the paper’s central claims. In particular, the authors support the claim that using the attention output space (via a combination of cross and self attention) produces sharper and more transferable saliency maps by demonstrating significant improvements in zero‐shot segmentation benchmarks (as shown in multiple tables and qualitative comparisons). The ablation studies further clarify that the combination of both attention types is crucial for the observed performance gains.

However, a couple of points could benefit from additional evidence:

• The claim that these representations are “highly interpretable” is mainly evaluated through segmentation metrics. Although improved segmentation performance is a strong indicator, a more in-depth human evaluation or analysis on other interpretability aspects could further substantiate this claim.

• The broader assertion regarding the transferability of DiT representations to other downstream tasks is demonstrated only in the context of segmentation. Additional experiments on diverse tasks would help confirm the generality of this transferability.

Overall, while the core experimental results are convincing, the claims about interpretability and broad transferability might be seen as slightly overreaching without further supporting evidence.

**Essential References Not Discussed:**

The paper is satisfactory, but a few additional references would help frame the contributions even better. For example:

• TCAV (Testing with Concept Activation Vectors by Kim et al., 2018) is a seminal work on concept-based interpretability. It shows how high-level concepts can be used to explain model decisions, which directly relates to the paper’s idea of using concept embeddings to generate saliency maps. Including a discussion of TCAV would help readers see how the current approach builds on or differs from established concept-based methods.

• The critique “Attention is not Explanation” by Jain and Wallace (2019) offers important context for any work that leverages attention mechanisms for interpretability. Although the authors argue that the attention output space in DiTs yields sharper and more reliable saliency maps, contrasting their findings with the limitations highlighted in that work would provide a more nuanced perspective.

Including these related works would better situate the paper’s contributions within the broader literature on interpretability and help clarify how its proposed method advances beyond previous approaches.

**Experimental Designs Or Analyses:**

I reviewed the experimental setups and analyses, and overall they appear sound and well-aligned with the paper’s objectives. For example:

- Segmentation Benchmarks:
  The use of standard zero-shot segmentation benchmarks (ImageNet-Segmentation and PascalVOC) to evaluate the quality of the saliency maps is appropriate. These benchmarks provide widely accepted metrics (mIoU, pixelwise accuracy, mAP) that serve as a robust proxy for assessing how well the method localizes textual concepts in images.

- Ablation Studies:
  The paper includes ablations that isolate the contributions of using just cross attention, just self attention, and their combination. This analysis clarifies that the integration of both mechanisms is key to achieving superior performance. Additionally, the study on the influence of diffusion timesteps helps understand how noise levels affect segmentation performance.

- Layer-wise Analysis:
  The experiments also examine the impact of using features from different layers of the model. This layered analysis is useful for demonstrating that deeper layers contribute more refined representations, and that aggregating information across layers further improves the results.

One potential concern is that while segmentation performance is an effective proxy for interpretability, it does not fully capture all aspects of what makes a model’s internal representations interpretable from a human perspective. A complementary human study or alternative qualitative analysis might have provided additional validation. However, given the context and common practices in this research area, the experimental designs and analyses are both reasonable and convincing.

**Methods And Evaluation Criteria:**

The proposed methods and evaluation criteria are well-aligned with the problem. The paper targets the challenge of interpreting diffusion models through precise, concept-specific saliency maps, and it does so by leveraging both cross and self attention within multi-modal DiTs. The use of standard zero-shot segmentation benchmarks like ImageNet-Segmentation and PascalVOC provides a robust framework to quantitatively and qualitatively assess the quality of these saliency maps. Additionally, the comprehensive ablation studies help confirm that the specific design choices, such as the combination of attention types, directly contribute to improved performance. This setup effectively demonstrates the method's utility for the intended application without introducing unnecessary complexity.

**Other Comments Or Suggestions:**

n/a

**Other Strengths And Weaknesses:**

Other Strengths:

- Originality:
  The paper creatively repurposes the attention mechanisms in multi-modal diffusion transformers to generate interpretable concept embeddings without requiring additional training. This inventive combination of ideas from diffusion models and attention-based interpretability represents a fresh perspective that advances the state of the art.

- Significance:
  By demonstrating state-of-the-art performance on zero-shot segmentation tasks, the work highlights the practical impact of its method. Its ability to produce sharp, transferable saliency maps not only deepens our understanding of DiT representations but also has potential implications for enhancing the transparency and controllability of generative models.

- Clarity:
  The paper is generally well-structured and clearly written, with detailed descriptions of the methodology, comprehensive experimental evaluations, and helpful pseudo-code that clarifies the proposed approach. The extensive ablation studies further reinforce the clarity of the experimental design and results.

Other Weaknesses:

- Generality:
  The method is demonstrated on multi-modal DiTs, and it remains somewhat unclear how well the approach would generalize to tasks beyond image segmentation. A discussion of these limitations could provide a more balanced perspective.

- Theoretical Underpinning:
  The paper could benefit from a deeper theoretical analysis of why the attention output space yields superior saliency maps compared to traditional cross-attention methods. While the empirical results are convincing, additional theoretical insights would enhance the overall robustness of the claims.

Overall, the paper makes a compelling contribution with its original approach and significant empirical findings, though further exploration in the areas noted above would provide additional depth and context to its contributions.

**Questions For Authors:**

1. Could you provide further insights—either theoretical or through additional experiments—on why the attention output space yields sharper and more transferable saliency maps compared to traditional cross-attention methods?

2. How sensitive is the method to the selection and number of concept tokens? For example, how does varying the vocabulary size or the choice of specific tokens affect the segmentation performance and interpretability?

3. Can you comment on the generalizability of CONCEPTATTENTION beyond multi-modal DiTs and the specific segmentation tasks evaluated? Have you explored or do you foresee its applicability to other architectures or downstream tasks?

4. Have you considered or conducted any human-centric evaluations of the interpretability provided by the saliency maps (e.g., user studies or qualitative assessments beyond segmentation metrics)?

5. Are there specific failure cases or limitations of CONCEPTATTENTION, particularly when dealing with images containing multiple overlapping or ambiguous objects?

**Relation To Broader Scientific Literature:**

The paper’s contributions build directly on and extend several strands of prior work in model interpretability, transformer architectures, and diffusion models. Specifically:

• Previous research has shown that attention mechanisms in models like UNet-based diffusion models can yield useful cross attention maps for localizing textual concepts (e.g., Tang et al., 2022). This work extends that idea by demonstrating that the output space of multi-modal diffusion transformers can be repurposed—using both cross and self attention—to generate even sharper, more transferable saliency maps.

• In the broader literature on transformer interpretability, methods such as GradCAM, Layer-wise Relevance Propagation, and Attention Rollout have been applied to vision transformers (including models like CLIP and DINO) to visualize and understand model decisions. The proposed CONCEPTATTENTION method builds on these insights by leveraging the rich, multi-modal representations inherent to diffusion transformers, thereby offering a new perspective on how internal representations can be made more interpretable.

• The paper also connects to recent work that explores how the representations of diffusion models can be utilized for downstream tasks such as segmentation. By showing that the same representations can be interpreted through concept embeddings to achieve state-of-the-art zero-shot segmentation performance, the paper bridges the gap between generative modeling and practical image analysis.

Overall, the work synthesizes ideas from transformer-based interpretability and diffusion model research, advancing the understanding of how multi-modal attention mechanisms can be manipulated to yield more precise and meaningful explanations of model behavior.

**Theoretical Claims:**

The paper does not include formal proofs for its theoretical claims. Instead, the authors provide algorithmic descriptions, equations, and intuitive justifications—such as the use of linear projections in the attention output space (e.g., Equation 13)—to support the conceptual basis of the method. The claim that this approach yields sharper and more transferable saliency maps is primarily validated through extensive empirical experiments and ablation studies, rather than through rigorous theoretical proofs. Consequently, there were no formal proofs to verify for correctness.

---

> ### Author Rebuttal · Authors · 2025-04-01
>
> Thank you for your insightful comments. After reading all of the reviews, we have implemented many of the requested experiments at [this anonymous website](https://concept-attention-anonymous.github.io/) and we will incorporate these updates into the camera ready paper. We are glad that the reviewers highlight the strengths of our work:
>
> 1. ConceptAttention is a **simple and elegant approach** (JHK8, XwNm) that produces **high quality saliency maps** with compelling empirical results (JHK8, XwNm, Njw6)
> 2. ConceptAttention **requires no additional training** (JHK8, XwNm, FGxX, Njw6)
> 3. and has the potential for **practical impact** to the community (JHK8, FGxX, Njw6) and is **well written and communicated** (JHK8, XwNm, FGxX, Njw6)
>
> We hope our responses below address your specific concerns.
>
> ---
> > a few additional references would help frame the contributions even better.
>
> We will absolutely include your suggeted references in the related works section of the final manuscript.
>
> > it remains somewhat unclear how well the approach would generalize to tasks beyond image segmentation. A discussion of these limitations could provide a more balanced perspective
>
> We actually found that ConceptAttention [generalizes seamlessly to video generation models](https://concept-attention-anonymous.github.io/#a)! We implemented ConceptAttention on the CogVideoX MMDiT video generation model and it generates qualitatively better saliency maps than the cross attention maps. Certainly, ConceptAttention has limitations, and we are happy to discuss more in the paper. For example, see our response to `2.` below.
>
> > 1. Could you provide further insights—either theoretical or through additional experiments—on why the attention output space yields sharper and more transferable saliency maps compared to traditional cross-attention methods?
>
> Great question! Textual information initially flows from the prompt tokens to the image patches. However, after the initial layers the image tokens themselves will encode the rich semantic information from the prompt. Cross attention only captures the *direct* contributions of text tokens to the image patches. Our approach captures both this information and the *indirect* semantic information flowing through the other image patches.
>
> > 2. How sensitive is the method to the selection and number of concept tokens?
>
> ConceptAttention is designed to pick the best concept for each patch out of those available, in much the same way that a zero-shot CLIP classifier would. This may lead to misattribution when there are very few concepts and none match the image contents. See the picture of a bike in [Fig F](https://concept-attention-anonymous.github.io/#f) for example. If the concepts “car” and “background” are chosen then “car” will be assigned to the bike as it is more similar than "background". However if both “car” and “bike” are given then the correct concept “bike” will be chosen.
>
> On the other hand, when there are many concepts and several have overlapping meanings, then ConceptAttention will still pick the one it decides is "best". This can result in one concept (i.e. "mountain") overpowering another, perhaps correct, concept like "tree". See [Fig G](https://concept-attention-anonymous.github.io/#g) for this example.
>
> > 3. Can you comment on the generalizability of ConceptAttention beyond multi-modal DiTs and the specific segmentation tasks evaluated? Do you foresee its applicability to other architectures or downstream tasks?
>
> As mentioned above, we found that ConceptAttention [generalizes seamlessly to video generation models](https://concept-attention-anonymous.github.io/#a). Additionally, we found it also generalizes to Stable Diffusion 3.5 Turbo, another T2I MMDiT model. We quantitatively evaluated ([see Table B](https://concept-attention-anonymous.github.io/#b)) it using the same protocol from Tab 1 in the manuscript and found it outperforms existing baselines. See [Fig C](https://concept-attention-anonymous.github.io/#c) for qualitative results.
>
> > 4. Have you considered or conducted any human-centric evaluations of the interpretability provided by the saliency maps (e.g., user studies or qualitative assessments beyond segmentation metrics)?
>
> A human-centric evaluation of our approach compared to other zero-shot interpretability methods would be a great line of future work. Of particular interest would be identifying if ConceptAttention can be used by non-experts to debug models, identifying why a model may not generate a proper image that aligns with the given prompt.
>
> > 5. Are there specific failure cases or limitations of ConceptAttention?
>
> Please see our answer to question 2 above.
>
> ---
>
> Once again, we thank the reviewer for their feedback and we hope our responses answered your remaining questions.

---

### Official Review · Reviewer_FGxX · 2025-03-10

**Overall Recommendation:** 3

**Summary:**

- This paper presents ConceptAttention, a method that leverages the attention of diffusion transformers (DiTs) to generate saliency maps for localizing textual concepts in images.
- By repurposing pre-trained  DiT's attention weights, the approach produces more accurate segmentation maps without requiring extra training.
- The work is timely, as DiTs are widespread, yet investigations to their attention were limited; this offers a fresh perspective with both scientific and practical impact.

**Claims And Evidence:**

- l.295-297: ‘However, these have a key limitation in that their vocabulary is limited to the tokens in the user’s prompt.’, I don’t think this is the right claim, as the concept is also the user’s prompt anyway, as stated in l.246-247.
- The claim made by the work provides a simple approach to visualising DiT attention without training and is, in general, innovative. There are sufficient visual results to support the claim. However, the quantitative evaluations are relatively weak, with unclear dataset specification and missing key related works to compare (specified in the following).

**Essential References Not Discussed:**

- The following key literature is missing:
    - OVAM [1] is highly relevant to the proposed method, as both share the same core architecture—using a parallel "concept prompt" to extract attention from a pre-trained diffusion model without training. The key difference is that [1] is implemented on a U-Net-based diffusion model, while ConceptAttention is based on DiT. Given this similarity, it is crucial to include [1] in both the related works section and the quantitative evaluation. Currently, the reported results in ConceptAttention’s Table 4 are not comparable to Table 2 in [1]. For example, DAAM achieves an mIoU of 66.2–79.7 in Table 2 of [1], whereas in Table 4 of ConceptAttention, it is only 10.97. While the specific PascalVOC subset used remains unclear, such a large discrepancy is unexpected.
    - CLIPasRNN [2], another training-free approach, should also be included in the evaluation. Specifically, zero-shot image segmentation results can be compared against Table 1 in [2].

[1] Marcos-Manchón, P., Alcover-Couso, R., SanMiguel, J.C. and Martínez, J.M., 2024. Open-vocabulary attention maps with token optimization for semantic segmentation in diffusion models. CVPR 2024.

[2] Sun, S., Li, R., Torr, P., Gu, X. and Li, S., 2024. Clip as rnn: Segment countless visual concepts without training endeavor. CVPR 2024.

**Experimental Designs Or Analyses:**

- To validate how the threshold impacts the proposed method, it would be helpful to plot the ROC curve comparing it with some key baseline methods (e.g. DAAM and Rollout CLIP).
- Important and closely related baseline methods need to be compared, e.g. OVAM [1] and CLIPasRNN [2].

**Methods And Evaluation Criteria:**

- The overall evaluation is on the right track, but some specifications need clarification:
    - The threshold used to generate segmentation masks from saliency maps is a crucial hyperparameter that significantly impacts results. However, this parameter is not reported in the methodology or experiments.
    - The multiclass evaluation is particularly questionable. The proposed method should naturally handle an arbitrary number of classes, yet the primary quantitative results focus on a simplified single-class setting (Table 1), with only limited results for the multiclass setting (Table 4).
    - The setup in Table 4 is unclear. For instance, how many classes from PascalVOC are included? Given the various PascalVOC versions, specifying these details is essential. Additionally, key baseline methods, such as OVAM [1] and CLIPasRNN [2], should be compared, as both provide PascalVOC results and share baseline models with this work (e.g., DAAM).

**Other Comments Or Suggestions:**

The main paper is well written, but additional details, such as the experimental setup, should be included in the appendix.

**Other Strengths And Weaknesses:**

The proposed approach, despite its similarity to [1], addresses a timely and important problem—investigating attention in DiT. The provided visual results sufficiently support this claim. The main concern lies in unclear details and missing quantitative results. Once these are clarified, I would be happy to reconsider my score.

**Questions For Authors:**

See above.

**Relation To Broader Scientific Literature:**

See below the ‘Essential References Not Discussed’ section.

**Theoretical Claims:**

- A key finding of ConceptAttention is that, in multi‐modal diffusion transformers (DiT), the prompt embeddings are dynamically updated alongside the image tokens, yet the concept tokens are designed to receive information from image tokens without feedback. This one‐way update mechanism allows the concept tokens to act as a semantic “anchor”—enabling the extraction of high-fidelity saliency maps that accurately localize textual concepts while preserving the image’s appearance. In contrast, U-Net–based diffusion models use static prompt embeddings, which simplifies visualization but lacks the flexible decoupling achieved in DiT.
- One key question arises: **Why is disabling the feedback from concept token to image token so important?** The author argued this is one key design innovation but doesn’t explain it in theory nor provide empirical ablation to support this claim.
- Another key question is, although the author attempts distinct concept tokens with prompt tokens, they are essentially the same (both from user-provided text and encoded with the same text encoder). The only difference is how they interact with image token (i.e. queation (9) and (10). **Therefore, my question is, what if the author just replaces the concept token with the prompt token and computes the saliency in the same way as concept attention**, i.g. $o_p = softmax(q_pk_{xp}^T)v_{xp}$?

---

> ### Author Rebuttal · Authors · 2025-04-01
>
> Thank you for your insightful comments. After reading all of the reviews, we have implemented many of the requested experiments at [this anonymous website](https://concept-attention-anonymous.github.io/) and we will incorporate these updates into the camera ready paper. We are glad that the reviewers highlight the strengths of our work:
>
> 1. ConceptAttention is a **simple and elegant approach** (JHK8, XwNm) that produces **high quality saliency maps** with compelling empirical results (JHK8, XwNm, Njw6)
> 2. ConceptAttention **requires no additional training** (JHK8, XwNm, FGxX, Njw6)
> 3. and has the potential for **practical impact** to the community (JHK8, FGxX, Njw6) and is **well written and communicated** (JHK8, XwNm, FGxX, Njw6)
>
> We hope our responses below address your specific concerns.
>
> ---
>
> > 'However, [existing models are] limited to the tokens in the user’s prompt.’, I don’t think this is the right claim as the concept is also the user’s prompt anyway
>
> Cross attention maps are by default restricted to the tokens in the user’s prompt. However, when generating images it is often desirable to segment concepts (e.g., "background") not explicitly in the prompt. We somehow need to add these new concepts to the prompt without impacting the generated image's appearance. Our method allows this.
>
> > Why is disabling the feedback from concept token to image token so important?
>
> ConceptAttention is a method for *interpreting the representations of MMDiT models during generation,* but we can’t use it as a tool for interpretation if our concepts change the image we are studying. Hence, we need to decouple concepts from image tokens.
>
> > [Can we just use the output vectors of prompt tokens?]
>
> Yep! Our first discovery was that the output space of MMDiT attention layers encode highly interpretable features. However, these maps are restricted to the prompt vocabulary (see previous answer). Our one-way attention flow removes this restriction.
>
> > The threshold used to generate segmentation masks [...] is not reported in the methodology or experiments.
>
> We hope to clarify this:
> 1. We choose the mean value of our saliency maps as the threshold. This choice was made to strictly adhere to the evaluation protocol laid out in (Chefer et al., CVPR 2021) and used in (Gandelsman et al., ICLR 2024), which both use the mean. We will more improve our description in the final paper.
> 2. We were also concerned that a particular choice of threshold could favor certain methods. Thus, we included the mean Average Precision (mAP) metric (Tab 1, 2, and 3) which is a *threshold agnostic metric* of segmentation performance that measures the weighted mean of precisions achieved across multiple thresholds.
>
> > The primary quantitative results focus on a simplified single-class setting (Tab 1), with only limited results for the multiclass setting (Table 4).
>
> Our experiments focus on a single-class setting to directly compare to the many zero-shot interpretability baselines which are only capable of generating single predictions. It would be unfair to expect methods like DINO to predict sensible maps for images with multiple classes. Of the subset of methods which can produce open vocabulary saliency maps (i.e., DAAM, TextSpan, and Cross Attention) our approach outperforms each of them.
>
> > Given the various PascalVOC versions, specifying these details is essential.
>
> Thank you for your feedback. We will include extensive experimental details in the appendix. Our single-class experiments cover all 20 classes in PascalVOC, but are restricted to 930 images with only one class present in them. Our multi-class experiments cover all 20 classes and all examples in the entire dataset of 1464 images, many containing multiple classes.
> > The reported results in ConceptAttention’s Tab 4 are not comparable to Tab 2 in OVAM (Marcos-Manchón et al., CVPR 2024).
>
> Tab 2 in the OVAM paper shows the evaluation of DAAM and OVAM on a synthetically generated dataset (introduced by the authors) called "VOC-sim", which is distinct from the VOC dataset we evaluate on. VOC-sim consists of images synthetically generated with prompts “a photograph of a {classname}” (Sec 4.1 of OVAM). This dataset is completely different from the VOC dataset we used.
>
> > It would be helpful to plot the ROC curve
>
> While not showing the ROC curve, the mean Average Precision (mAP) metric does capture the area under the precision-recall curve.
>
> > Baseline methods need to be compared [OVAM and CLIPasRNN]
>
> Following the reviewer's suggestions we implemented OVAM and CLIPasRNN as additional baselines. We found that our method outperforms both of these (see [Tab B](https://concept-attention-anonymous.github.io/#b)).
>
> ---
>
> Thanks again! If our responses above are satisfactory, we would greatly appreciate the reviewer increasing their score to reflect their increased confidence in our work.

---

### Official Review · Reviewer_XwNm · 2025-03-11

**Overall Recommendation:** 4

**Summary:**

The paper introduces ConceptAttention, a novel method for generating saliency maps based on user-defined textual concepts. These maps are of high quality and achieve state-of-the-art performance on zero-shot image segmentation benchmarks, surpassing other interpretability methods. Notably, ConceptAttention does not require any retraining and is easy to understand. This demonstrates that the features of multi-modal diffusion transformers (MMDiTs) are highly transferable and potentially beneficial for various downstream vision tasks.

**Claims And Evidence:**

Claims are well supported by experiments.

**Essential References Not Discussed:**

1) DINOv2: Learning Robust Visual Features without Supervision, TMLR 2024: Although comparison to DINO features were explored in the paper, it would be nice to also compare to the latest version of DINO.
2) Vision Transformers Need Registers, ICLR 2024: This work improves the DINOv2 features even further. Again, it would be great to see the performance against this method.

**Experimental Designs Or Analyses:**

Yes, experimental designs and analysis are good. Ablation studies are interesting and insightful.

**Methods And Evaluation Criteria:**

Yes.

**Other Comments Or Suggestions:**

1) In Section 4.1, in equations (4), (5), and (6), there are in total $k$ concepts which I believe should be $r$ instead as it was mentioned in the beginning of the paragraph: "The user specifies a set of $r$ single token concepts...".

**Other Strengths And Weaknesses:**

Strengths:
1) The paper is well written and easy to read.
2) The idea is simple and effective.
3) The claims are well supported empirically.
4) The proposed method is interesting from the interpretability perspective and potentially useful in downstream vision tasks.

Weaknesses:
1) Some missing references and comparisons (please see above).
2) Limited model evaluation, i.e., only Flux-Schnell model was validated so far. It would be great to see the method performance with other MMDiTs too.

**Questions For Authors:**

1) How about other MMDiT models except Flux-Schnell (e.g., Stable Diffusion 3)? Is the performance of the method equally good?
2) Will the method work with usual DiT-based models (e.g., PixArt family of models)? Is there a way to make it work with this architecture?
3) I also wonder about video generation DiT models? Do you think it is possible to extend your method to them? What kind of information can be extracted from there?

**Relation To Broader Scientific Literature:**

N/A

**Theoretical Claims:**

N/A

---

> ### Author Rebuttal · Authors · 2025-04-01
>
> We thank the reviewer for their insightful comments. After reading all of the reviews, we have implemented many of the requested experiments which you can see at this [anonymous website](https://concept-attention-anonymous.github.io/). We are glad to see that multiple reviewers recognize the strengths of our work:
>
> 1. ConceptAttention is a **simple and elegant approach** (JHK8, XwNm) that produces **high quality saliency maps** with compelling empirical results (JHK8, XwNm, Njw6)
> 2. ConceptAttention **requires no additional training** (JHK8, XwNm, FGxX, Njw6)
> 3. and has the potential for **practical impact** to the community (JHK8, FGxX, Njw6) and is **well written and communicated** (JHK8, XwNm, FGxX, Njw6)
>
> The major changes of particular relevance to reviewer XwNm are:
>
> 1. **ConceptAttention works on a Video Generation model!** Watch the demo [here](https://concept-attention-anonymous.github.io/#a).
> 2. **ConceptAttention generalizes to Stable Diffusion 3.5 Turbo.** Quantitative results are shown in [Fig B](https://concept-attention-anonymous.github.io/#b) and qualitative results are shown in [Fig C](https://concept-attention-anonymous.github.io/#c).
> 3. **ConceptAttention is now compared against additional baselines.**
> Following reviewer suggestions, we implemented: DINOv2, DINOv2 with registers, iBOT, OVAM, and CLIP as RNN and found that our method outperforms each of them. [See Table B](https://concept-attention-anonymous.github.io/#b).
>
> These updated results will all be incorporated into our camera ready paper. Below we aim to address the particular concerns of reviewer XwNm.
>
> ---
>
> > Although comparison to DINO features were explored in the paper, it would be nice to also compare to the latest version of DINO. `[...]` This work improves the DINOv2 features even further.
>
> The reviewer raises an interesting request for comparison against DINOv2 (Oquab et al., TMLR 2024) and DINOv2 with Registers (Darcet et al., ICLR 2024), both of which are highly relevant. Sec. 5 of our original submission evaluated the performance of the self-attention maps from both of these methods, and we found that ConceptAttention outperforms both of them.
>
> Our further analysis showed that DINOv2 actually has less interpretable self-attention maps than DINOv1, despite performing better on downstream tasks. See a table summarizing these results in [Table B](https://concept-attention-anonymous.github.io/#b). Surprisingly, we also found that DINOv2 with registers under-performed compared to DINOv2. We also provided qualitative results which subjectively match the self-attention results shown in each respective paper in [Fig D](https://concept-attention-anonymous.github.io/#d).
>
> > 1. How about other MMDiT models except Flux-Schnell (e.g., Stable Diffusion 3)? Is the performance of the method equally good?
>
> Yes! We followed your suggestion and implemented our approach on a Stable Diffusion 3.5 Turbo model and found that it produces competitive results on the same quantitative evaluation we conducted for Table 1 of the manuscript. ConceptAttention on SD3.5 Turbo beats all tested baselines on both ImageNet-Segmentation and PascalVOC, though ConceptAttention on the Flux-Schnell architecture is slightly better than it on most metrics. See the [Table B]() for quantitative results and [Fig C]() for qualitative results.
>
> > 2. Will the method work with usual DiT-based models (e.g., PixArt family of models)? Is there a way to make it work with this architecture?
>
> Our approach hinges upon MMDiT models that leverage multi-modal attention layers that jointly process both text and image modalities, and thus will not work with the T2I paradigm of e.g., the PixArt family.
>
> > 3. Do you think it is possible to extend your method to [video generation models]?
>
> Yes! To answer your question, we implemented ConceptAttention on CogVideoX (Yang et. al, ICLR 2025) and found that our approach seamlessly generalizes to video generation models. The only difference is that we also average information over the temporal dimension. We found that ConceptAttention produces qualitatively better results than cross attention maps from the same model. See the [video demonstration](https://concept-attention-anonymous.github.io/#a).
>
> ---
>
> Once again, we thank the reviewer for their feedback. If our responses and new results are satisfactory, we would greatly appreciate the reviewer increasing their score to reflect their increased confidence in our work.

---

### Official Review · Reviewer_JHK8 · 2025-03-16

**Overall Recommendation:** 4

**Summary:**

The authors present a new method to extract well-refined saliency maps from pre-trained DiT models without having to perform any additional training, mainly by directly leveraging the attention weights of the multi-modal model in a clever way to establish correspondences to a set of provided ‘concepts’ that might appear in the image – providing a neat approach for improved (layer-wise) insights into these blackbox models.

**Claims And Evidence:**

The main claims made during the early stages of the paper are all well substantiated through experimental evidence.
The only critical point I see is the claim that ConceptAttention has “minimal impact on model latency” (l 217 right) – as this is only true for a small set of concept embeddings; As this set is included in a self-attention operation, larger sets will inevitably cause larger latencies due to the quadratic complexity of this operation!

**Essential References Not Discussed:**

None that come to mind in direct relation to the work's main contributions;
Potential updates to Table 1 could be DinoV2 (Oquab et al., TMLR 2024) or iBOT (Zhou et al., ICLR2022)

**Experimental Designs Or Analyses:**

The comparative baselines in Fig. 5 and Table 1 might slightly be skewed to the authors’ advantage, e.g. the choice of DINO instead of the much-newer and more powerful DINOv2, or other alternatives that commonly return better saliency maps (e.g. iBOT, …);

As mentioned previously:
In Section 5.1, the choice of the threshold as the *mean value* to produce binary segmentation masks for the quantitative analysis is an understandable but potentially suboptimal choice that could distort the results – the mean is, after all, highly affected by outliers/extreme values;
$\textrightarrow$ Choice of median, and/or top-x % as a cut-off might be more reliable (see questions);

Minor: Additional ablation regarding the choice of a simple dot-product to produce saliency maps could be interesting to justify this choice (L 258 right).

**Methods And Evaluation Criteria:**

The task of zero-shot image segmentation as main basis for the evaluation of the ‘concept maps’ is well-chosen, as it is a reasonable way to quantify the object-specific salience maps;

The comparative baselines might be slightly skewed to the authors’ advantage, see ‘experimental designs’ section below.

The choice of the threshold as the mean value to produce binary segmentation masks for the quantitative analysis is an understandable but potentially suboptimal choice that could distort the results – the mean is, after all, highly affected by outliers/extreme values; Choice of median, and/or top-x % as a cut-off might be more reliable (see questions);

**Other Comments Or Suggestions:**

I’d suggest the authors go through their manuscript in detail and correct the typos / grammatical mistakes, e.g.
- L 142 right: “a diffusion models” ($\textrightarrow$ model)
- L 173 left: “line of work attempts perform” ($\textrightarrow$ attempts to or performs)
- L 241 right: “at the end of attention operation” ($\textrightarrow$ end of the attention ..)
- L 265 right: $\textrightarrow$ Should start upper-case after period: This is..
- L 412 right: “pixewlise” ($\textrightarrow$ pixelwise)
- L 414 right: “out performed” ($\textrightarrow$ outperformed)
- …

**Other Strengths And Weaknesses:**

**Strengths**:
*Originality & Significance:*
- The authors provide a simple and neat but powerful approach which yields high-quality saliency maps and allows a variety of query-concepts to be tested for, hence provides a good measure of flexibility on this axis
- The authors’ method repurposes the already trained parameters of the underlying multi-modal DiT model, which entirely removes any need for additional training and/or fine-tuning – providing great benefit to the community

*Clarity:*
- The paper is generally well written and easy to follow, with a good number of clear visualisations (e.g. Figure 4) supporting the contributions and explanations

---
**Weaknesses**:

- Missing discussion of cases where concepts are queries that are in fact NOT in the image – see questions.
- Missing discussion and/or analysis of behaviour dependent on number of concept queries, as well as potential partial overlap of provided concepts – see questions
- Quantitative evaluation might be skewed towards certain methods that have a clearer separation around the mean – which is the threshold the authors choose; This could be improved upon by additionally evaluating using the median or top-X% -- see questions;
- Quality of the manuscript should be improved – there are several typos and grammatical errors that can (and should) easily be corrected (see comments)
- Minor: Comparative methods (Table 1) are mostly pre-2022, e.g. DINO v1 instead of the more powerful v2; see questions

**Questions For Authors:**

*TLDR;* I do like the approach, as I think it is a very simple and elegant yet powerful method to provide insights! However, I'd like the authors to address a number of questions! Depending on the responses, I'm happy to update my rating!

**Major:**
1. What happens if concepts are provided as a query that are NOT contained in the image? I’d be curious to hear/see whether the model will be able to recognise their non-existance, or still pick out irrelevant areas as a saliency map! And are other concepts’ saliency maps negatively affected?
2. How does the quality of the saliency maps change if more or fewer concepts are provided? Is there a ‘sweet-spot’ in terms of number of concepts? What happens if overlapping concepts are provided, e.g. “landscape” and “mountain”/”grass”?
3. As previously mentioned, I feel like the mean as the threshold to create the binary decision / saliency map might skew the results towards methods that don’t produce outliers (which is, of course, still a valid choice). However: have the authors investigated how median as a metric, as well as smaller top-X% (e.g. top 30%) would change this?


**Others:**

4.  Fig 6: is the ‘combined’ information from all layers, or the layers 10-18 as detailed previously in the experimental setup description?
Independent of this, why do the authors think the combined approach outperforms all individual layers? How exactly are the layers combined? I am slightly surprised about this result, since I’d expect e.g. the average to lie somewhere in-between the extremes; Or are the individual contributions simply making it more robust in terms of the threshold metric?
5.  Although not necessarily in direct competition with the proposed approach, Table 1 does list methods like DINO – however, there have been significant improvements after 2021 in terms of DinoV2 as well as other methods like iBOT, which have shown to often produce better saliency maps; I’d be good to see some results for these methods as well if possible to get a feeling how well their saliency maps perform (Note: I don’t expect your method to be better, but it would just provide more up-to-date insight to the reader!)

More of a suggestion:

6.  I feel like the “*Impact of diffusion timestep on segmentation*” section in Section 5.2 would deserve more highlighting! It is quite interesting to see that the middle diffusion timesteps perform significantly better than both early and late ones!
Do the authors have more intuitions why this could be?
7.  I think it would be interesting to the reader to include a visualisation, i.e. qualitative analysis, how the saliency maps progress across different layers throughout – in addition to the quantitative plot that’s currently shown.

---
---
## Post-Rebuttal Update:
Given that the authors have sufficiently addressed all my concerns and have provided many additional convincing insights, I am raising my score from 2 to 4 and recommend acceptance of this work.

**Relation To Broader Scientific Literature:**

Relation to broader literature is sufficient; The authors also discuss their constraints in terms of not comparing to methods trained on large datasets like SAM.

**Theoretical Claims:**

No theoretical claims beyond well-known/established formulas.

---

> ### Author Rebuttal · Authors · 2025-04-01
>
> Thank you for your thorough response. After reading all of the reviews, we have implemented many of the requested experiments at [this anonymous website](https://concept-attention-anonymous.github.io/) and we will incorporate these updates into the camera ready paper. We are glad that the reviewers highlight the strengths of our work:
>
> 1. ConceptAttention is a **simple and elegant approach** (JHK8, XwNm) that produces **high quality saliency maps** with compelling empirical results (JHK8, XwNm, Njw6)
> 2. ConceptAttention **requires no additional training** (JHK8, XwNm, FGxX, Njw6)
> 3. and has the potential for **practical impact** to the community (JHK8, FGxX, Njw6) and is **well written and communicated** (JHK8, XwNm, FGxX, Njw6)
>
> We hope our responses below address your specific concerns.
>
> ---
>
> > Only critical point I see is the claim that ConceptAttention has “minimal impact on model latency”
>
> We agree this is imprecise wording. This statement indeed holds only when $c << p$ (for $c$ concepts and $p$ patches). Thankfully, for typical values of $c$ (i.e., 1, 10, 50), the patch self-attention operations $O(p^2)$ dominate the number concept attention operations $O(pc)$.  ConceptAttention on a single NVIDIA A40 for 1,5, and 50 concepts the model takes 1.12, 1.14, and 1.20 seconds respectively to perform a forward pass.
> > 1. What happens if concepts are provided as a query that are NOT contained in the image?
>
> This is a great question! ConceptAttention is designed to pick the most relevant concept out of those given, in the same way that a zero-shot CLIP classifier would. This means if the most similar concept out of those given is incorrect then it may still be chosen.
> For example, take an image of a bike on the street, if the concepts “car” and “background” are chosen then “car” will likely be assigned to the bike as it is more similar than "background". However if both “car” and “bike” are given then the correct concept “bike” will be chosen ([see Fig F](https://concept-attention-anonymous.github.io/#f)).
> > 2. How does the quality of the saliency maps change if more or fewer concepts are provided? [...] What happens if overlapping concepts are provided?
>
> ConceptAttention picks the best concept for each patch out of those available. This may lead to misattribution when there are very few concepts and none match the image contents (see previous answer). However, when there are many concepts and several have similar or overlapping meanings, then ConceptAttention will still emphasize just one. This can result in one concept (i.e, "mountain") overpowering another valid concept like "tree" ([see Fig G](https://concept-attention-anonymous.github.io/#g)).
> > 3. The choice of the threshold as the mean value [is] potentially suboptimal
>
> Thank you for the opportunity to clarify our choice of threshold:
> 1. The mean value was chosen in an effort to strictly adhere to the evaluation protocol laid out in (Chefer et al., CVPR 2021, also used by (Gandelsman et al., ICLR 2024) which uses the mean.
> 2. To prevent a particular choice of threshold favoring certain methods, we included the mean Average Precision (mAP) metric (Table 1, 2, 3) which is a *threshold agnostic metric* measuring segmentation performance that takes the weighted mean of precisions achieved across multiple thresholds.
>
> > 4. Fig 6: is the ‘combined’ information from all layers?
>
> We collect concept and image embeddings from each of these layers, compute their projections, and then average over the layer dimension. This improves robustness to noise from individual layers.
> > 5. Potential updates to Tab 1 could be DINOv2 or iBOT.
>
> Thank you for the suggestion. We implemented iBOT, DINOv2 and DINOv2 with registers. We found that our approach outperformed each of them quantitatively on the same evaluation shown in Table 1. Intriguingly, the raw self-attention maps of DINOv2 underperform compared to the DINOv1 model. An example of maps from each of these methods is [shown in Fig D](https://concept-attention-anonymous.github.io/#d).
> > 6. Do the authors have intuitions for why [the middle steps are better than late ones]?
>
> This was an interesting result to us as well. We have observed that early steps early steps shape the semantic, high-level structure of an image (with too much noise for high quality segmentation maps) while later steps focus on high-frequency minor details. Thus, the middle steps likely offer a good balance between these two extremes.
> > 7. It would be interesting to [show] how the saliency maps progress across different layers
>
> This is a great suggestion! See [Fig E](https://concept-attention-anonymous.github.io/#e) for these results, which align with Fig 6 from the paper.
>
> ---
>
> Thanks again for your feedback! If our responses and new results are satisfactory, we would greatly appreciate the reviewer increasing their score to reflect their increased confidence in our work.

---

> > ### Comment · Reviewer_JHK8 · 2025-04-05
> >
> > I'd like to congratulate the authors on the additional insights they have provided in the rebuttal, which (in my opinion) make the paper significantly stronger.
> >
> > All my queries have been sufficiently addressed; I also couldn't spot any other prohibitive weaknesses when reading through the other reviews -- and hence, I'm updating my rating to recommend acceptance.

---

### Decision · Program_Chairs · 2025-05-01

**Decision:**

Accept (oral)

**Comment:**

The paper proposes a novel method for extracting well-refined saliency maps from pretrained DIT models in a training-free manner by leveraging attention weights with an additional set of concept tokens. It received four reviews, and after a successful rebuttal, the final ratings are three accepts and one weak accept.

All reviewers acknowledge the novelty and effectiveness of the method, the soundness of the evaluation design, and the clarity of the presentation. Overall, in line with the reviewers, the AC believes this work addresses a timely and important problem and is likely to attract interest from the community.